# Universal Function Approximation on Graphs

**Rickard Brüel-Gabrielsson**
`rbg@cs.stanford.edu`

## Abstract

In this work we produce a framework for constructing universal function approximators on graph isomorphism classes. We prove how this framework comes with a collection of theoretically desirable properties and enables novel analysis. We show how this allows us to achieve state-of-the-art performance on four different well-known datasets in graph classification and separate classes of graphs that other graph-learning methods cannot. Our approach is inspired by persistent homology, dependency parsing for NLP, and multivalued functions. The complexity of the underlying algorithm is $O(\#\text{edges} \times \#\text{nodes})$ and code is publicly available[1].

## 1 Introduction

Graphs are natural structures for many sources of data, including molecular, social, biological, and financial networks. Graph learning consists loosely of learning functions from the set of graph isomorphism classes to the set of real numbers, and such functions include node classification, link prediction, and graph classification. Learning on graphs demands effective representation, usually in vector form, and different approaches include graph kernels [10], deep learning [24], and persistent homology [1]. Recently there has been a growing interest in understanding the discriminative power of certain frameworks [22, 7, 13, 12] which belongs to the inquiry into what functions on graph isomorphism classes can be learned. We call this the problem of function approximation on graphs. In machine learning, the problem of using neural networks (NNs) for function approximation on $\mathbb{R}^d$ is well-studied and the universal function approximation abilities of NNs as well as recurrent NNs (RNNs) are well known [11, 18]. In this work, we propose a theoretical foundation for universal function approximation on graphs, and in Section 3 we present an algorithm with universal function approximation abilities on graphs. This paper will focus on the case of graph classification, but with minor modifications, our framework can be extended to other tasks of interest. We take care to develop a framework that is applicable to real-world graph learning problems and in Section 4 we show our framework performing at state-of-the-art on graph classification on four well known datasets and discriminating between graphs that other graph learning frameworks cannot.

Among deep learning approaches, a popular method is the graph neural network (GNN) [23] which can be as discriminative as the Weisfeiler-Lehman graph isomorphism test [22]. In addition, Long Short Term Memory models (LSTMs) that are prevalent in Natural Language Processing (NLP) have been used on graphs [20]. Using persistent homology features for graph classification [9] also show promising results. Our work borrows ideas from persistent homology [8] and tree-LSTMs [21].

To be able to discriminate between any isomorphism classes, graph representation should be an injective function on such classes. In practice this is challenging. Even the best known runtime [4] for such functions is too slow for most real world machine learning problems and their resulting representation is unlikely to be conducive to learning. To our knowledge, there exists no algorithm that produces isomorphism-injective graph representation for machine learning applications. We overcome several challenges by considering multivalued functions, with certain injective properties, on graph isomorphism classes instead of injective functions.

Our main contributions: (i) Showing that graph representation with certain injective properties is sufficient for universal function approximation on bounded graphs and restricted universal function approximation on unbounded graphs. (ii) A novel algorithm for learning on graphs with universal function approximation properties, that allows for novel analysis, and that achieves state-of-the-art performance on four well known datasets. Our main results are stated and discussed in the main paper, while proof details are found in the Appendix.

## 2 Theory

An overview of this section: (i) Multivalued functions, with injective properties, on graph isomorphism classes behave similarly to injective functions on the same domain. (ii) Such functions are sufficient for universal function approximation on bounded graphs, and (iii) for restricted universal function approximation on unbounded graphs. (iv) We postulate what representation of graphs that is conducive to learning. (v) We relate universal function approximation on graphs to the isomorphism problem, graph canonization, and discuss how basic knowledge about these problems affects the problem of applied universal function approximation on graphs. (vi) We present the outline of an algorithmic idea to address the above investigation.

### 2.1 Preliminaries

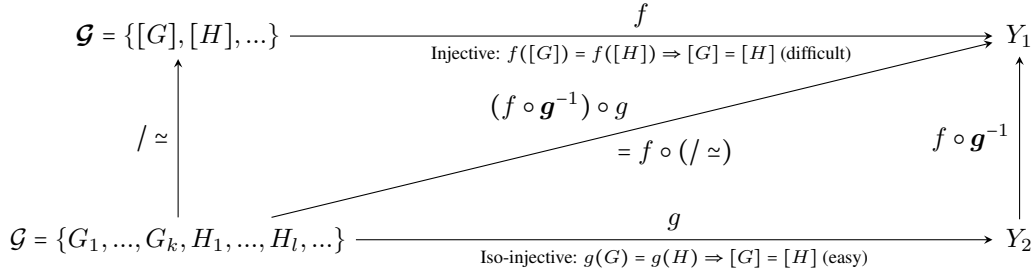

Figure 1: Diagram of the relations between injective functions on graph isomorphism classes, $\boldsymbol{\mathcal{G}}$, and iso-injective functions on graphs, $\mathcal{G}$. Constructing iso-injective functions on $\mathcal{G}$ is much easier than constructing injective functions on $\boldsymbol{\mathcal{G}}$, and by the existence of the well-defined function $f \circ \boldsymbol{g}^{-1}$ we do not lose much by switching our attention to iso-injective functions on $\mathcal{G}$.

**Definition 1.** A *graph* (undirected multigraph) $G$ is an ordered triple $G := (V(G), E(G), l)$ with $V(G) := \{1, 2, \ldots, n\}$ a set of vertices or nodes, $E(G)$, a multiset of $m$ unordered pairs of nodes, called edges, and a label function $l : V(G) \to \mathbb{N}_+$ on its set of nodes. The *size of graph* $G$ is $|G| := |V(G)| + |E(G)| + \sup\{l(v) \mid v \in V(G)\}$, and we assume all graphs are finite.

**Definition 2.** Two graphs $G$ and $H$ are *isomorphic* ($G \simeq H$) if there exists a bijection $\phi : V(G) \to V(H)$ that preserves edges and labels, i.e. a graph *isomorphism.*

**Definition 3.** Let $\mathcal{G}$ denote the set of all finite graphs. For $b \in \mathbb{N}$ let $\mathcal{G}_b \subset \mathcal{G}$ denote the set of graphs whose size is bounded by $b$.

**Definition 4.** Let $\boldsymbol{\mathcal{G}}$ denote the set of all finite graph isomorphism classes, i.e. the quotient space $\mathcal{G}/\simeq$. For $b \in \mathbb{N}$ let $\boldsymbol{\mathcal{G}}_b \subset \boldsymbol{\mathcal{G}}$ denote the set of graph isomorphism classes whose size is bounded by $b$, i.e. $\mathcal{G}_b/\simeq$. In addition, we denote the graph isomorphism class of a graph $G \in \mathcal{G}$ as $[G]$ (coset) meaning for any graphs $G, H \in \mathcal{G}$, $[G] = [H]$ if and only if $G \simeq H$.

**Lemma 1.** *The sets $\mathcal{G}$ and $\boldsymbol{\mathcal{G}}$ are countably infinite, and the sets $\mathcal{G}_b$ and $\boldsymbol{\mathcal{G}}_b$ are finite.*

**Definition 5.** A function $f : \mathcal{G} \to Y$ is *iso-injective* if it is injective with respect to graph isomorphism classes $\boldsymbol{\mathcal{G}}$, i.e. for $G, H \in \mathcal{G}$, $f(G) = f(H)$, implies $G \simeq H$.

**Definition 6.** A *multivalued function* $f : X \Rightarrow Y$ is a function $f : X \to \mathcal{P}(Y)$, i.e. from $X$ to the powerset of $Y$, such that $f(x)$ is non-empty for every $x \in X$.

**Definition 7.** Any function $f : \mathcal{G} \to Y$ can be seen as a multivalued function $\boldsymbol{f} : \boldsymbol{\mathcal{G}} \Rightarrow Y$ defined as $\boldsymbol{f}([G]) := \{f(H) \mid H \in [G]\}$ and we call the size of the set $\boldsymbol{f}([G])$ the *class-redundancy* of graph isomorphism class $[G]$.

Let $Alg : \mathcal{G} \to \mathbb{R}^d$ be an iso-injective function. For a graph $G \in \mathcal{G}$ we call the output of $Alg(G)$ the *encoding* of graph $G$. The idea is to construct a universal function approximator by using the universal function approximation properties of NNs. We achieve this by composing $Alg$ with NNs and constructing $Alg$ itself using NNs. Without something similar to an injective function $f : \mathcal{G} \to Y$ we will not arrive at a universal function approximator on $\mathcal{G}$. However, we do not lose much by using a multivalued function $\boldsymbol{g} : \mathcal{G} \Rightarrow Y$ that corresponds to an iso-injective function $g : \mathcal{G} \to Y$.

**Theorem 1.** *For any injective function $f : \mathcal{G} \to Y$ and iso-injective function $g : \mathcal{G} \to Y$ there is a well-defined function $h : \mathrm{im}(g) \to Y$ such that $f = h \circ g$.*

See Figure 1 for a diagram relating these different concepts. For completeness, we also add the following theorem.

**Theorem 2** (recurrent universal approximation theorem [18]). *For any recursively computable function $f : \{0, 1\}^* \to \{0, 1\}^*$ there is a RNN $\phi$ that computes $f$ with a certain runtime $r(|w|)$ where $w$ is the input sequence.*

Unfortunately Theorem 2 requires a variable number of recurrent applications that is a function of the input length, which can be hard to allow or control. Furthermore, the sets of graphs we analyze are countable. This makes for a special situation, since a lot of previous work focuses on NNs' ability to approximate Lebesgue integrable functions, but countable subsets of $\mathbb{R}$ have measure zero, rendering such results uninformative. Thus, we focus on pointwise convergence.

## 2.2 Bounded Graphs

With an iso-injective function, universal function approximation on bounded graphs is straightforward.

**Theorem 3** (finite universal approximation theorem). *For any continuous function $f$ on a finite subset $X$ of $\mathbb{R}^d$, there is a NN $\varphi$ with a finite number of hidden layers containing a finite number $n$ of neurons that under mild assumptions on the activation function can approximate $f$ perfectly, i.e. $\|f - \varphi\|_\infty = \sup_{x \in X} |f(x) - \varphi(x)| = 0$.*

From Theorem 1 and since $\mathcal{G}_b$ is finite we arrive at the following:

**Theorem 4.** *Any function $f : \mathcal{G}_b \to \mathbb{R}$ can be perfectly approximated by any iso-injective function $Alg : \mathcal{G}_b \to \mathbb{R}^d$ composed with a NN $\varphi : \mathbb{R}^d \to \mathbb{R}$.*

## 2.3 Unbounded Graphs

For a function to be pointwise approximated by a NN, boundedness of the function and its domain is essential. Indeed, in the Appendix we prove (i) there is no finite NN with bounded or piecewise-linear activation function that can pointwise approximate an unbounded continuous function on an open bounded domain, and (ii) there is no finite NN with an activation function $\sigma$ and $k \geq 0$ such that $\frac{d^k \sigma}{dx^k} = 0$ that can pointwise approximate all continuous functions on unbounded domains.

**Theorem 5** (universal approximation theorem [11]). *For any $\epsilon > 0$ and continuous function $f$ on a compact subset $X$ of $\mathbb{R}^d$ there is a NN $\varphi$ with a single hidden layer containing a finite number $n$ of neurons that under mild assumptions on the activation function can approximate $f$, i.e. $\|f - \varphi\|_\infty = \sup_{x \in X} |f(x) - \varphi(x)| < \epsilon$.*

Though universal approximation theorems come in different forms, we use Theorem 5 as a ballpark of what NNs are capable off. As shown above, continuity and boundedness of functions are prerequisites. This forces us to take into account the topology of graphs. Indeed, any function $f : \mathcal{G} \to \mathbb{R}^d$ with a *bounded* co-domain will have a convergent subsequence for each sequence in $\mathcal{G}$, by Bolzano-Weierstrass. Since a NN $\varphi : \mathbb{R}^d \to \mathbb{R}^d$ may only approximate continuous functions on $\mathrm{im}(f)$, the same subsequences will be convergent under $\varphi \circ f$. Thus, since $\mathcal{G}$ is countably infinite and due to limiting function approximation abilities of NNs, we always, for any $f$, have a convergent *infinite* sequence *without repetition* of graph isomorphism classes. Furthermore, $f$ *determines* such convergent sequences independent of $\varphi$ and should therefore be learnable and flexible so that the convergent sequences can be adapted to the specific task at hand. See Appendix for more details. This leads to the following remark:

**Remark 1.** An injective function $f : \mathcal{G} \to \mathbb{R}^d$ determines a non-empty set of convergent *infinite* sequences *without repetition* in $\mathcal{G}$ under the composition $g = \varphi \circ f$ with any NN $\varphi$. Meaning that $f$ affects which functions $g$ can approximate. Thus, for flexible learning, $f$ should be flexible and learnable to maximize the set of functions that can be approximated by $g$. Hopefully then, we can learn an $f$ such that two graphs $[G]$ and $[H]$ that are close in $\|f([G]) - f([H])\|$ are also close according to some useful metric on $\mathcal{G}$. The same holds for iso-injective functions $Alg : \mathcal{G} \to \mathbb{R}^d$.

We are left to create a function $Alg : \mathcal{G} \to \mathbb{R}^d$ that is bounded but we cannot guarantee it will be closed so that we may use Theorem 5; however, we add this tweak:

**Theorem 6.** *For any $\epsilon > 0$ and bounded continuous function $f$ on a bounded subset $X$ of $\mathbb{R}^d$ there is a NN $\varphi$ with a single hidden layer containing a finite number $n$ of neurons that under mild assumptions on the activation function can approximate $f$, i.e. $\|f - \varphi\|_\infty = \sup_{x \in X} |f(x) - \varphi(x)| < \epsilon$.*

For example, we can bound any iso-injective function $Alg : \mathcal{G} \to \mathbb{R}^d$ by composing (this simply forces the convergent sequences to be the values in $\mathbb{R}^d$ with increasing norm) with the injective and continuous Sigmoid function $\sigma(x) = \frac{1}{1+e^{-x}}$.

## 2.4 Learning and Graph Isomorphism Problems

**Definition 8.** The *graph isomorphism problem* consists in determining whether two finite graphs are isomorphic, and *graph canonization* consists in finding, for graph $G$, a canonical form $Can(G)$, such that every graph that is isomorphic to $G$ has the same canonical form as $G$.

The universal approximation theorems say nothing about the ability to learn functions through gradient descent or generalize to unseen data. Furthermore, a class of graphs occurring in a learning task likely contains non-isomorphic graphs. Therefore, to direct our efforts, we need a hypothesis about what makes learning on graphs tractable.

**Postulate 1.** *A representation (encoding) of graphs that facilitates the detection of shared subgraphs (motifs) between graphs is conducive to learning functions on graphs.*

With this in mind, an ideal algorithm produces for each graph a representation consisting of the multiset of canonical forms for all subgraphs of the graph. Even better if the canonical representations of each graph are close (for some useful metric) if they share many isomorphic subgraphs. However, there is a few challenges: (i) The fastest known algorithm for the graph canonization problem runs in quasipolynomial $2^{O((\log n)^c)}$ time [4], and (ii) a graph has exponentially $\Omega(n!)$ many distinct subgraphs.

First, obtaining a canonical form of a graph is expensive and there is no guarantee that two graphs with many shared subgraphs will be close in this representation. Second, obtaining a canonical form for each subgraph of a graph is even more ungainly. We approach these challenges by only producing iso-injective encodings of a graph and a sample of its subgraphs. Iso-injective encodings of graphs are easily obtained in polynomial time. However, we still want small class-redundancy and flexibility in learning the encodings.

## 2.5 Algorithmic Idea

We construct a *universal function approximator on graph isomorphism classes of finite size* by constructing a multi-set of encodings that are *iso-injective*. Ideally, for efficiency, an algorithm when run on a graph $G$ constructs iso-injective encodings for subgraphs of $G$ as a subprocess in its construction of an iso-injective encoding of $G$. Thus, a recursive local-to-global algorithm is a promising candidate. Consider Algorithm 1; the essence of subset parsing is the following:

**Theorem 7.** *For Algorithm 1 the encoding $c(S_{1,2})$ with $S_{1,2} = S_1 \cup S_2$ and $|V(S_{1,2})| + |E(S_{1,2})| = p > 1$ is iso-injective if we have on input graph $G$*

> *1. for all $S \in A \subset G$, with $|V(S)| + |E(S)| < p$*
>
> > *(a) the encoding $c(S)$ is iso-injective*
> > *(b) each label $l(v)$ for $v \in V(S)$ is unique*
>
> *2. $r$ is an injective function*

---

**Algorithm 1** Subset Parsing Algorithm

---

   **Input:** Graph $G$,
      set $A$ of subgraphs of $G$, and functions $c : A \to \mathbb{R}^{d_c}, \quad r : \{\mathbb{R}^{d_c}, \mathbb{R}^{d_c}\} \times \mathcal{P}(h(V)) \times \mathbb{N} \to \mathbb{R}^{d_c}$
   **Output:** Extended function $c : A \to \mathbb{R}^{d_c}$
   **for** $S_1, S_2 \in A$ **do**
      Let $S_{1,2} = S_1 \cup S_2$
      $c(S_{1,2}) = r(\{c(S_1), c(S_2)\}, \{l(v) \mid v \in V(S_1) \cap V(S_2)\}, |V(S_{1,2})| + |E(S_{1,2})|)$
      $A = A \cup \{S_{1,2}\}$
   **end for**

---

We envision an algorithm that combines encodings of subgraphs $S_1, \ldots, S_n$ into an encoding of graph $S_{1,\ldots,n}$, such that if $c(S_1), \ldots, c(S_n)$ are iso-injective so is $c(S_{1,\ldots,n})$. However, we need to make sure all labels are unique within each subgraph and to injectively encode pairwise intersections.

## 3 Method

Methods such as GNNs successfully aggregate label and edge information in a local-to-global fashion; however, GNNs lack sufficiently unique node identification to extract fully expressive representations [22]. The quickly growing number (unbounded for graphs in $\mathcal{G}$) of intersections in GNNs' processing of subgraphs complicates analysis. Our method keeps processed subgraphs disjoint (Lemma 2) which allows for comparatively simple inductional analysis. We ensure that within a processed subgraph each node-encoding is unique, which together with some additional properties proves sufficient to produce iso-injective encodings for graphs (Theorem 9). Parsing disjoint subgraphs by adding one edge at a time is inspired by 0-dimensional persistent homology [8]; the idea being that our method may revert to computing 0-dimensional persistence based on increasing node degrees, and should therefore (neglecting overfitting) perform no worse than certain persistence based kernels [1, 9]. See Figure 2 for how message (or information) passing occurs in Node Parsing (Algorithm 2) versus in GNNs.

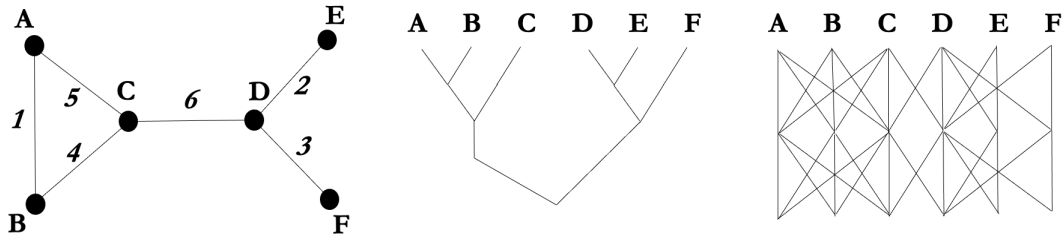

Figure 2: Left to right: Graph with edge-ordering. Message passing in Node Parsing on graph. Message passing in GNN on same graph.

In this section we present Algorithm 2 and show how with the use of NNs it is a *universal function approximator on graphs* (Theorem 10). This section is outlined as follows: (i) A description of the Node Parsing Algorithm (NPA). (ii) Proving that, under certain requirements on the functions that NPA make use of, NPA produces iso-injective representations of graphs. (iii) Proving the existence of functions with the prerequisite requirements. (iv) Proving NNs can approximate such functions. (v) Presenting a weaker baseline model for comparison. (vi) Analysis of class-redundancy, parallelizability, and introducing the concept of subgraph droupout.

### 3.1 The Algorithm

**Lemma 2.** *In Algorithm 2, an edge (in the second for loop) is always between two disjoint subgraphs in $A_i$ or within the same (with respect to =) subgraph in $A_i$. Also, each subgraph in $A_i$ is disjoint and connected.*

---

**Algorithm 2** Node Parsing Algorithm (NPA)

---

**Input:** Graph $G$,

functions $s_e : E \times \mathcal{G} \to \mathbb{R}$, $s_v : V \times \mathcal{G} \to \mathbb{R}$, $h_{init} : \mathbb{N}_+ \to \mathbb{R}^{d_v}$, $c_{init} : \mathbb{R}^{d_v} \to \mathbb{R}^{d_c - 1}$

functions $r_c : \mathbb{R}^{2d_c + 2d_v} \times \{0,1\} \to \mathbb{R}^{d_c}$, $r_v : \mathbb{R}^{d_c + d_v} \times \{0,1\} \to \mathbb{R}^{d_v}$,

special symbol $zero$

**Output:** Multisets $W(G) := [w_i \mid i = 1, \ldots, n+m]$, $C(G) := [c(S) \mid S \in A_{m+1}] \subset W(G)$

Let $A_1 = V(G)$ // Where each node is seen as a subgraph of $G$

**for** $i = 1, \ldots, n$ **do**

$\quad h^1(v_i) = h_{init}(l(v_i))$

$\quad w_i = c(v_i) = c_{init}(h^1(v_i))$.append($zero$) // step 0 encode

**end for**

Sort $E$ with $s_e(\cdot, G)$ so $s_e(e_1, G), \ldots, s_e(e_m, G)$ are in ascending order

**for** $i = 1, \ldots, m$ **do**

$\quad$ Let $(v_a, v_b) = e_i$ and sort $(v_a, v_b)$ ascendingly with $s_v(\cdot, G)$

$\quad$ Let $S_1, S_2 \in A_i$ be subgraphs with $v_a \in S_1$ and $v_b \in S_2$

$\quad$ Let $S_{1,2} = S_1 \cup S_2 \cup (v_a, v_b)$

$\quad w_{n+i} = c(S_{1,2}) = r_c(\{(c(S_1), h^i(v_a)), (c(S_2), h^i(v_b))\}, \mathbb{1}_{S_1 = S_2})$ // step $i$ encoding of $S_{1,2}$

$\quad h^{i+1} = h^i$ // inheriting previous $h$-values

$\quad$ **for** $v \in V(S_{1,2})$ **do**

$\quad \quad h^{i+1}(v) = r_v(c(S_{1,2}), h^i(v), \mathbb{1}_{v \in V(S_1)})$;

$\quad$ **end for**

$\quad A_{i+1} = (A_i - \{S_1, S_2\}) \cup \{S_{1,2}\}$

**end for**

---

**Theorem 8.** *For Algorithm 2, each produced $c$-encoding is iso-injective, if $h_{init}$, $c_{init}$, and $r_c$ are injective, if for all subgraphs $S_1, S_2 \in A_i$ that appear at step $i$ when run on input graph $G$*

- *each value $r_v(c(S_{1,2}), \tilde{h}, \mathbb{1}_{v \in V(S_1)})$ for $\tilde{h} \in h^i(V(S_1) \cup V(S_2))$ is unique,*

*and if for all graphs $S_{1,2}, S_{1,2}^*$ with $c := c(S_{1,2}) = c(S_{1,2}^*)$, encoded at step $i$ run $G$ and step $j$ run $H$ respectively,*

- *$r_v(c, \cdot, \mathbb{1}_{v \in V(S_1)})$ is injective across $\{h^i(v) \mid v \in V(S_{1,2})\}$ and $\{h^j(v) \mid v \in V(S_{1,2}^*)\}$*

By Lemma 2, intersection is encoded by $\mathbb{1}_{S_1 = S_2}$ and uniqueness of $h$-values is established by properties of $r_v$ (specifically, $\mathbb{1}_{S_1 = S_2}$ allows us to discern whether a new edge is between two disjoint isomorphic subgraphs, with identical $c$-encodings, or within the same subgraph). Thus, the proof follows almost immediately from Theorem 7. Furthermore, and critically, $r_v(c(S_{1,2}), \cdot, \mathbb{1}_{v \in V(S_1)})$ being injective across $\{h^i(v) \mid v \in V(S_{1,2})\}$ and $\{h^j(v) \mid v \in V(S_{1,2}^*)\}$ ensures that if we find that two graphs are isomorphic after having applied $r_v$ they were also isomorphic before the application of $r_v$, all the way back to the original node-labels. The special $zero$-symbol allows us to assert whether an encoded graph has zero edges, as we otherwise want to deconstruct an encoded subgraph by considering two earlier encoded subgraphs connected by an edge.

### 3.2 Existence of Required Functions

In providing functions with the prerequisite properties we rely on the fact that our labels live in $\mathbb{N}_+$. This is necessary since we want to be able to use NNs, which can only approximate continuous functions, while at the same time our method injectively compresses label and connectivity information. In particular, there exists a continuous and bounded function from $\mathbb{R}^2$ to $\mathbb{R}$ that is injective in $\mathbb{N}^2$, while there exists no continuous function from $\mathbb{R}^2$ to $\mathbb{R}$ that is injective in $\mathbb{R}^2$.

Suppose the $c$-encoding of a subgraph $S_k$ consists of $c(S_k) = (y_k, m_k^1, m_k^2)$ and consider functions

$$h_{init}(l(v)) = l(v) \in \mathbb{N}_+, \quad c_{init}(h) = (0, 0, h+1)$$

and for subgraphs $S_1$ and $S_2$ with $S_{1,2} = S_1 \cup S_2 \cup (v_a, v_b)$

$$c(S_{1,2}) := r_c(\{(c(S_1), h(v_a)), (c(S_2), h(v_b))\}, \mathbb{1}_{S_1 = S_2}) =$$
$$\left(r(\{(y_1, h(v_a), m_1^1, m_1^2), (y_2, h(v_b), m_2^1, m_2^2)\}, \mathbb{1}_{S_1 = S_2}), m_1^2 + m_2^2 + 1, 2m_1^2 + 2m_2^2 + 2\right)$$
$$r_v(c(S_{1,2}), h(v), \mathbb{1}_{v \in V(S_1)}) = \left\{ \begin{array}{ll} h(v) + m_{1,2}^1, & \text{if } \mathbb{1}_{v \in V(S_1)} = 1 \\ h(v), & \text{else} \end{array} \right\}$$

where

$$\tau(i,j) = \frac{(i+j)(i+j+1)}{2} + j, \quad \rho(i,j) = (i+j, ij)$$
$$r(y_1, h_1, m_1, n_1, y_2, h_2, m_2, n_2, b) = \tau\left(\tau\left(\rho(\tau^4(y_1, h_1, m_1, n_1), \tau^4(y_2, h_2, m_2, n_2))\right), b\right)$$

In the Appendix we prove that the functions presented in this section satisfy the requirements in Theorem 8, which allows us to arrive at the following:

**Theorem 9** (NPA Existence Theorem). *There exists functions for Algorithm 2 such that every produced graph encoding is iso-injective.*

### 3.3 Corollaries

In our discussion of Algorithm 2 we will assume that it uses functions such that Theorem 9 holds. See Appendix for additional corollaries and remarks.

**Corollary 1.** *For Algorithm 2, given graphs $G, H \in \mathcal{G}$, $G \simeq H$ if and only if $C([G]) \cap C([H]) \neq \varnothing$. I.e. it solves the graph isomorphism problem and canonization.*

**Corollary 2.** *For graphs $G, H \in \mathcal{G}$ consider multiset $I = W(G) \cap W(H)$. Each $w \in I$ corresponds to a shared subgraph between $G$ and $H$, and $|I|$ is a lower bound to the number of shared subgraphs. The graph corresponding to $I$ is a lower bound (by inclusion) to the largest shared subgraph.*

**Lemma 3.** *Assume $\mathcal{X}$ is countable. There exists a function $f : \mathcal{X} \to \mathbb{R}^n$ so that $h(X) = \sum_{x \in X} f(x)$ is unique for each multiset $X \subset \mathcal{X}$ of bounded size. Moreover, any multiset function $g$ can be decomposed as $g(X) = \phi(\sum_{x \in X} f(x))$ for some function $\phi$.*

**Corollary 3.** *If $\mathcal{G}_* \subset \mathcal{G}$ and $\{|C(G)| \mid G \in \mathcal{G}_*\}$ is bounded (number of connected components is bounded), there exists a function $f$ such that any two graphs $G$ and $H$ in $\mathcal{G}_*$ are isomorphic if $\sum_{c \in C(G)} f(c) = \sum_{c \in C(H)} f(c)$.*

In the Appendix we show, given a graph isomorphism class $[S]$ and using NPA, a Turing-decidable function for detecting the presence of $[S]$ within a graph $G$; however, if we only have one global encoding for all of $G$ such a Turing-decidable function might not exist. Unless there is some subgraph-information in the encoding we are left to enumerate an infinite set, which is Turing-undecidable. This points to the strength of having the encoding of a graph $G$ coupled with encodings of its subgraphs.

### 3.4 Use of Neural Networks

**Theorem 10** (NPA Universal Approximation Theorem). *Functions $r_v, r_c, h_{init}, c_{init}$ that satisfies requirements of Theorem 8, and a function $f_3$ enabling Lemma 3 from Section 3.3, can be perfectly approximated by NNs for graphs in $\mathcal{G}_b$ and pointwise approximated for graphs in $\mathcal{G}$.*

By Theorem 3, NNs can perfectly approximate any function on a finite domain so the case of $\mathcal{G}_b$ is straightforward. However, for countably infinite $\mathcal{G}$ the situation is different. Consider functions from Section 3.2 and 3.3 (Lemma 3). They are continuous (in $\mathbb{R}^*$) but not bounded, we are applying these functions recursively and would want both the domain and the image to be bounded iteratively. Without losing any required properties we can compose these functions with an injective, bounded, and continuous function with continuous inverse such as Sigmoid, $\sigma$, and use $h_{init}(l(v)) = \sigma(l(v))$. Then these functions can be pointwise approximated by NNs. However, recursive application of a NN might increase the approximation error. We use NNs for all non-sort functions. For $r_c$ we use a tree-LSTM [21] and for $r_v$ we use a LSTM. See Appendix for details.

### 3.5 A Baseline

To gauge how conducive our approach is to learning and how important the strict isomorphic properties are, we present a simpler and non iso-injective baseline model which is the same as Algorithm 2 but the second outer for-loop has been replaced by Algorithm 3. Some results of this algorithm can be seen in Table 1 and it performs at state-of-the-art.

---

**Algorithm 3** Node Parsing Baseline Algorithm (NPBA)

---

    **for** i=1, ..., m **do**
        Let $(v_a, v_b) = e_i$ and let $S_1, S_2 \in A_i$ be subgraphs with $v_a \in S_1$ and $v_b \in S_2$
        $c(S_{1,2}) = r_c(\{c(S_1), c(S_2)\})$
    **end for**

---

### 3.6 Class-Redundancy, Sorting, Parallelize, and Subgraph Dropout

The class-redundancy in the algorithm and functions we propose enters at the sort functions $s_e$ (sorts edges) and $s_v$ (sorts nodes within edges). Thus, a loose upper bound on the class-redundancy is $O((m!)2^m)$. A better upper bound is $O((t_{1,1}!) \dots (t_{1,l_1}!)(t_{2,1}!) \dots (t_{k,l_k}!)(2^p))$ where each $t_{i,j}$ is the number of ties within group $j$ of groups of subgraphs that could be connected within the tie $i$. The order in between disconnected tied subgraph groups does not affect the output. See Appendix for #edge-orders, i.e. $O((t_{1,1}!) \dots (t_{1,l_1}!)(t_{2,1}!) \dots (t_{k,l_k}!))$, on some datasets.

We focus on function $s_e$. Each edge can be represented by the following vector [*deg1*, *deg2*, *label1*, *label2*]. We assume *deg1*, *deg2* as well as *label1*, *label2* are in descending order, and that ties are broken randomly. This work makes use of four $s_e$ functions: (i) *none*: Does not sort at all. (ii) *one-deg*: Sorts by *deg1*. (iii) *two-degs*: Sorts lexicographically by *deg1*, *deg2*. (iv) *degs-and-labels*: Sorts lexicographically by *deg1*, *deg2*, *label1*, *label2*.

Since the encodings of subgraphs that share no subgraph do not affect each other, we can parellalize our algorithm to encode such subgraphs in parallel. For example, a graph of just ten disconnected edges can be parellalized to run in one step. We call the number of such parellalizable steps for a graph's *levels*. See Appendix for #levels on some datasets.

In most cases, one run of NPA on graph $G$ computes features for a very small portion of all subgraphs of $G$. We could run NPA on all possible orders to make sure it sees all subgraphs, but this is very costly. Instead, we use the random sample of featurized subgraphs as a type of dropout [19]. During training, at each run of the algorithm we use only one ordering of the edges, which discourages co-adaptation between features for different subgraphs. At testing, we let the algorithm run on a sample of $K$ orderings, and then average over all these runs. We call this technique *subgraph dropout*.

## 4 Experiments

See Table 1 for results on graph classification benchmarks. We report average and standard deviation of validation accuracies across the 10 folds within the cross-validation. In the experiments, the $W(G)$ features are summed and passed to a classifier consisting of fully connected NNs. For NPA, $s_v$ sorts randomly, but with "-S", $s_v$ sorts based on the levels of subgraphs $S_1$ and $S_2$. For subgraph dropout "-D" we use $K = 5$. The four bottom rows of Table 1 compare different functions for sorting edges ($s_e$).

### 4.1 Synthetic Graphs

We showcase synthetic datasets where the most powerful GNNs are unable to classify the graphs, but NPA is. See Appendix for related discussion and Table 2 where

1. GNN-Hard: Class 1: Two disconnected cycle-graphs of $n/2$ vertices. Class 2: One single cycle-graph of $n$ vertices. ($n = 2, 4, 6, \dots, 32$)
2. NPBA-Hard: Class 1: Two nodes with $m$ edges in between. Class 2: Two nodes, with $m$ self-edges from one of the nodes. ($m = 2, 3, 4, \dots, 19$)
3. Erdos: Random Erdos-Renyi graphs.

Table 1: GNN is best performing variant from [22]. *: Best result with and without subgraph dropout.

| Datasets: | NCI1 | MUTAG | PROTEINS | PTC |
|---|---|---|---|---|
| # graphs: | 4110 | 188 | 1113 | 344 |
| # classes: | 2 | 2 | 2 | 2 |
| PatchySan [16] | 78.6±1.9 | 92.6±4.2 | 75.9±2.8 | 60.0±4.8 |
| DCNN [3] | 62.6 | 67.0 | 61.3 | 56.6 |
| DGCNN [14] | 74.4±4.7 | 85.8±1.6 | 75.5±0.9 | 58.6±2.5 |
| GNN [22] | 82.7±1.7 | 90.0±8.8 | 76.2±2.8 | 66.6±6.9 |
| NPBA (ours) | 81.0±1.1 | 92.8±6.6 | 76.6±5.7 | 67.1±5.9 |
| NPBA-D (ours) | 83.7±1.5 | 92.2±7.9 | **77.1±5.3** | 65.5±6.8 |
| NPA (ours) | 81.8±1.9 | 92.8±7.0 | 76.9±3.0 | **67.6±5.9** |
| NPA-D (ours) | **84.0±2.2** | 92.8±7.5 | 76.8±4.1 | 67.1±6.9 |
| NPA-S (ours) | 81.5±1.6 | **93.3±6.0** | 76.5±5.0 | 65.9±8.3 |
| NPA-D-S (ours) | 83.0±1.2 | **93.3±6.0** | 76.3±4.5 | 66.2±7.7 |
| NPA* (degs-and-labels) | 83.2±1.6 | 88.9±10.5 | 75.9±5.4 | 63.2±6.3 |
| NPA* (two-degs) | **84.0±2.2** | 91.7±6.7 | 76.2±4.6 | **67.6±5.9** |
| NPA* (one-deg) | 79.2±1.9 | 92.8±7.0 | 76.5±4.9 | 64.7±7.0 |
| NPA* (none) | 77.7±3.0 | 92.8±7.5 | 76.9±3.0 | 65.3±5.9 |

Table 2: (Train-accuarcy). Comparing NPA against other methods for certain types of graphs.

| Datasets: | GNN-Hard | NPBA-Hard | Erdos | Erdos-Labels | Random-Regular |
|---|---|---|---|---|---|
| # graphs: | 32 | 36 | 30 | 100 | 10 |
| # classes: | 2 | 2 | 30 | 100 | 10 |
| Avg # nodes: | 17±9 | 1.5±0.5 | 10±0 | 10±0 | 8±0 |
| Avg # edges: | 34±19 | 21±10 | 45±7 | 45±7 | 16±0 |
| $O$(median # edge-orders): | $10^{35}$ | $10^{21}$ | $10^{19}$ | $10^{8}$ | $10^{10}$ |
| GNN (GIN) [22] | 50 | **100** | **100** | **100** | 10 |
| NPBA (ours) | **100** | 50 | 83 | **100** | 70 |
| NPA (ours) | **100** | **100** | **100** | **100** | **90** |

4. Random-Regular: Each node has the same degree with configuration model from [15].

# 5 Discussion

In this paper, we develop theory and a practical algorithm for universal function approximation on graphs. Our framework is, to our knowledge, theoretically closest to a universal function approximator on graphs that performs at the state-of-the-art on real world datasets. It is also markedly different from other established methods and presents new perspectives such as subgraph dropout. In practice, our framework shares weaknesses with GNNs on regular graphs, and we do not scale as well as some other methods. Future work may reduce the class-redundancy, explore bounds on expected class-redundancy, modify GNNs to imbue them with iso-injective properties, or combine iso-injective encodings (from NPA) with invariant encodings (from GNNs) to enable the best of both worlds.

# 6 Broader Impact

This work helps advance the fields of machine learning and AI, which as a whole is likely to have both positive and negative societal consequences [17, 5]; many of which might be unintended [6]. The coupling of application and theory in this work aims at improving human understanding of AI which is related to efforts within for example explainable AI [2]. Such efforts may reduce unintended consequences of AI.

# 7 Acknowledgements

This work was supported by Altor Equity Partners AB through Unbox AI (www.unboxai.org). I am grateful for Bradley J. Nelson's help in reading the paper and for his suggestions on how to make it clearer. I also want to express my greatest gratitude to Professor Gunnar Carlsson and Professor Leonidas Guibas for their unwavering support and belief in me.

## Footnotes

[1] `https://github.com/bruel-gabrielsson/universal-function-approximation-on-graphs`

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
