[Supplementary Material]

# Appendices

## A  Theory

### A.1  Preliminaries: Additional Definitions, Remarks, and Proofs

#### A.1.1  Additional Definitions and Remarks

We add the following definitions:

**Definition 9.** A *subgraph* $S$ of a graph $G$, denoted $S \subset G$, is another graph formed from a subset of the vertices and edges of $G$. The vertex subset must include all endpoints of the edge subset, but may also include additional vertices.

**Definition 10.** We denote the disjoint union between two sets $A, B$ as $A \sqcup B$.

**Definition 11.** We denote the set-builder notation for multisets as $[x \mid Predicate(x)]$, i.e. with brackets to emphasize it constructs a multi-set.

**Definition 12.** If we write $f(A)$ where $A$ is a subset of the domain of $f$, we mean the multiset $f(A) := [f(x) \mid x \in A]$.

**Definition 13.** Let $f : X \to Y$ be a function from a set $X$ to a set $Y$. If a set $A$ is a subset of $X$, then the restriction of $f$ to $A$ is the function

$$f|_A : A \to Y$$

given by $f|_A(x) = f(x)$ for $x$ in $A$. Informally, the restriction of $f$ to $A$ is the same function as $f$, but is only defined on $A \cap dom(f)$.

**Definition 14.** For an iso-injective function $f : \mathcal{G} \to Y$ we define the *iso-inverse* as the function $\boldsymbol{f}^{-1} : \text{im}(f) \to \mathcal{G}$, where $\text{im}(f) = \{y \mid y \in Y, \exists G \in \mathcal{G}, f(G) = y\}$, as

$$\boldsymbol{f}^{-1}(y) = [G], \exists G \in \mathcal{G}, f(G) = y$$

**Definition 15.** The *subgraph isomorphism problem* consists in, given two graphs $G$ and $H$, determining whether $G$ contains a subgraph that is isomorphic to $H$.

**Definition 16.** With a function $f : X \to Y$ being injective across domains $X_1$ and $X_2$ with $X_1, X_2 \subset X$, we mean that for all $x_1 \in X_1, x_2 \in X_2$ with $f(x_1) = f(x_2)$ we have $x_1 = x_2$.

**Definition 17.** In some proofs we say *subgraph $S$ encoded at step $j$* of Algorithm 2 (NPA), with which we mean that if $j = 0$ then $S$ is a single node that is encoded in the first for loop of NPA, and if $j > 0$ then $S$ contains an edge and is encoded in the second for loop of NPA with $j = i$.

We also add the following remarks:

**Remark 2.** Functions on nodes $f : V(G) \to Y$, such as node labels, are functions of graphs too, because it makes no sense to compare indices or nodes between different graphs that are not subgraphs of the same graph. That is, each such function is different for each graph $G$, so if we abuse notation when having also a graph $H$ and $f : V(H) \to Y$ in a shared context with $G$, then $v_1 = v_2$ implies $f(v_1) = f(v_2)$ only if $v_1, v_2 \in V(G)$ or $v_1, v_2 \in V(H)$. Similarly, intersection between edges or nodes of two graphs $S_1$ and $S_2$ is only interesting to us if $S_1, S_2$ are subgraphs of some graph $G$.

**Remark 3.** We can bound any iso-injective function $Alg : \mathcal{G} \to \mathbb{R}^d$ by composing (this simply forces the convergent subsequence to be the values in $\mathbb{R}^d$ with increasing norm) with the injective and continuous Sigmoid function $\sigma(x) = \frac{1}{1+e^x}$.

#### A.1.2  Proof of Lemma 1

*Proof.* For each $n \in \mathbb{N}_+$ there is a finite number of graphs $G$ with $|V(G)| + |E(G)| + \sup_{v \in V(G)}(l(v)) = n$, and a countable union of countable sets is countable. Similarly, bounded graphs means that such a $n$ is bounded by $b$, and a finite union of finite sets is finite. Furthermore, $|\boldsymbol{\mathcal{G}}| \leq |\mathcal{G}|$ and $|\boldsymbol{\mathcal{G}}_b| \leq |\mathcal{G}_b|$. $\square$

#### A.1.3  Proof of Theorem 1

*Proof.* Consider, $h = (f \circ \boldsymbol{g}^{-1}) : \mathcal{G} \to Y$ which is well defined since $\boldsymbol{g}^{-1}$ is a function on $\text{im}(g)$, and $f = h \circ g$. $\square$

### A.1.4 Proof of Theorem 2

*Proof.* See [20] for proof. □

## A.2 Bounded Graphs

### A.2.1 Proof of Theorem 3

*Proof.* In [2] it is proven that any continuous piecewise linear function is representable by a ReLU NN, and any finite function can be perfectly approximated by a continuous piecewise linear function. □

### A.2.2 Proof of Theorem 4

*Proof.* Consider the function $g : \mathrm{im}(Alg) \to \mathbb{R}^d$:

$$g(x) = (f \circ \boldsymbol{Alg}^{-1})(x)$$

Which is well-defined because both $f$ and $\boldsymbol{Alg}^{-1}$ are functions on their respective domains. Since $\mathrm{im}(Alg)$ is a finite subset of $\mathbb{R}^d$ we know there is a NN $\varphi$ that perfectly approximates $g$, and thus we have

$$f = \varphi \circ Alg$$

□

## A.3 Unbounded Graphs

## A.4 On Remark 1

Suppose $Alg : \mathcal{G} \to \mathbb{R}^d$ is an iso-injective function and $\varphi : \mathbb{R}^d \to \mathbb{R}$ is a NN. We analyze the functions $f : \boldsymbol{\mathcal{G}} \to \mathbb{R}$ that $\varphi \circ Alg$ can approximate. By Theorem 5, if $\mathrm{im}(Alg) \subset \mathbb{R}^d$ is bounded, then $\varphi$ can approximate all continuous functions on the closure $\overline{\mathrm{im}(Alg)}$. Since $\mathcal{G}$ is countably infinite, we may consider the sequence $\mathrm{im}(Alg) = (\boldsymbol{Alg}([G]_i)_{j=0}^{k_i})_{i=0}^{\infty} = ((a_i)_{j=0}^{k_i})_{i=0}^{\infty} \subset \mathbb{R}^d$. From the Bolzano-Weierstrass Theorem we know *every bounded sequence of real numbers has a convergent subsequence*. If $\mathrm{im}(Alg)$ is bounded then so is $((a_i)_{j=0}^{k_i})_{i=0}^{\infty}$, and thus it has a convergent subsequence. Similarly, the subsequence $Alg([G]_{i=0}^{\infty})$ with $Alg([G]_i) = Alg(H), H \in [G]_i$, corresponding to a sequence over the graph isomorphism classes $[G]_i \in \boldsymbol{\mathcal{G}}$, has a convergent subsequence. Meaning that for every $\delta > 0$ there is a countably infinite set $A \subset \boldsymbol{\mathcal{G}}$ such that $[G]_i, [G]_j \in A$ implies $\|Alg([G]_i) - Alg([G]_j)\| < \delta$. Let $L$ denote the limit point of one such convergent subsequence. By Theorem 5, we assume that $\varphi$ can approximate only continuous functions, this means for every $\epsilon > 0$ there exists a $\delta > 0$ such that that $\|L - Alg([G])\| < \delta$ with $[G] \in \boldsymbol{\mathcal{G}}$ implies $\|\varphi(L) - \varphi(Alg([G]))\| < \epsilon$. Note that the same holds for an injective function $h : \boldsymbol{\mathcal{G}} \to \mathbb{R}^d$, because the sequences $\mathrm{im}(h) = h([G]_{i=0}^{\infty})$ and $((a_i)_{j=0}^{k_i})_{i=0}^{\infty}$ have the same cardinality.

## A.5 Theorems and Proofs

**Theorem 11.** *There is no finite width and depth NN with bounded or piecewise-linear activation function that can pointwise approximate an unbounded continuous function on an open bounded domain.*

*Proof.* Such NNs must be bounded on bounded domains. □

**Theorem 12.** *There is no finite width and depth NN with an activation function $\sigma$ and $k \geq 0$ such that $\frac{d^k \sigma}{dx^k} = 0$ that can pointwise approximate all continuous functions on unbounded domains.*

*Proof.* Consider $f(x) = x^{k+1}$ such that $\frac{d^k f}{x^k} \neq 0$. The NN cannot asymptotically approximate $f$. □

**Theorem 13** (Bolzano-Weierstrass). *Every bounded sequence of real numbers has a convergent subsequence.*

*Proof.* Well-known result, see Wikipedia or your favorite analysis book. □

### A.5.1 Proof of Theorem 5

*Proof.* Proof can be found in [21] and [8] for a large family of activation functions. □

### A.5.2 Proof of Theorem 6

*Proof.* If $X$ is closed, it follows immediately from Theorem 5. Suppose $X$ is open, then we know by Theorem 5 that $\varphi$ can pointwise approximate $f$ on a compact set, but since $f$ is bounded we know that each limit point is finite. Thus, we may just add them and define $g$ as $f$ extended with the limit points. Then $g$ is continuous on a compact $\overline{X}$, so $\varphi$ pointwise approximates $g$, but this means it also pointwise approximates $f$. □

## A.6 Algorithmic Idea

### A.6.1 Proof of Theorem 7

*Proof.* Suppose Algorithm 1 is run on graphs $G$ and $G^*$. Suppose also that the assumptions of the theorem holds for both runs and that $c(S_{1,2}) = c(S_{1,2}^*)$ with $S_{1,2} \subset G, S_{1,2}^* \subset G^*$. This means, since $p > 1$ that we can split up in the following way, $S_{1,2} = S_1 \cup S_2$ with $S_1, S_2 \in A \subset G$ and $S_{1,2}^* = S_1^* \cup S_2^*$ with $S_1^*, S_2^* \in A^* \subset G^*$. We want to show that $S_{1,2} \simeq S_{1,2}^*$.

We know since $r$ is injective that

$$c(S_1) = c(S_1^*), \ c(S_2) = c(S_2^*), \tag{1}$$

$$\{l(v) \mid v \in V(S_1) \cap V(S_2)\} = \{l(v) \mid v \in V(S_1^*) \cap V(S_2^*)\} \tag{2}$$

(If instead $c(S_1) = c(S_2^*), c(S_2) = c(S_1^*)$ we can just relabel) This means that there exists isomorphisms $\phi_1 : S_1 \to S_1^*$ and $\phi_2 : S_2 \to S_2^*$.

Consider the following map:

$$\phi(v) = \begin{cases} \phi_1(v) & \text{if } v \in V(S_1) \\ \phi_2(v) & \text{otherwise} \end{cases} \tag{3}$$

We set $I = V(S_1) \cap V(S_2)$. Now, since both $\phi_1$ and $\phi_2$ are isomorphisms we know that $\phi$ respects $l$-values, and the only part of the domain where $\phi$ might not respect edges is in $I$. Now let $I^* = V(S_1^*) \cap V(S_2^*)$.

All values in $l(I)$ are unique among $l(V(S_1) \cup V(S_2))$, all values in $l(I^*)$ are unique among $l(V(S_1^*) \cup V(S_2^*))$. From Equation 2 we know that $l(I) = l(I^*)$. Suppose $v \in I$ then $\phi_1(v) = \phi_2(v)$ because else $l(\phi_1(v)) \neq l(\phi_2(v)) \to l(v) \neq l(v)$ by the stated uniqueness of the $l$-values of $I$ and $I^*$. Since, $\phi_1$ and $\phi_2$ agree on the intersection $I$ we know that all edges must be respected by $\phi$ by construction.

Now we want to show that $\phi$ is a bijection. From construction we know that $\phi$ is a bijection on $V(S_1) \to V(S_1^*)$. Now $V(S_1) \cup V(S_2) = V(S_1) \sqcup (V(S_2) - I)$ and $V(S_1^*) \cup V(S_2^*) = V(S_1^*) \sqcup (V(S_2^*) - I^*)$. From before we know that $\phi(I) = I^*$. Thus, we know that $\phi$ is injective map on $V(S_2) - I \to B \subset V(S_2^*) - I^*$ because $\phi$ is equivalent to $\phi_2$ on that domain. To see this, suppose $v \in V(S_2) - I$ and $\phi(v) \in V(S_1^*)$, then we must have $\phi(v) \in I^*$ (since $\phi(v) = \phi_2(v) \in V(S_2^*)$), but this would mean that $v \in I$ (else $l$-value cannot be respected by uniqueness) and we would get a contradiction. Lastly, since $|V(S_2) - I| = |V(S_2)| - |I|, |V(S_2)| = |V(S_2^*)|, |I| = |I^*|$, and $|V(S_2^*) - I^*| = |V(S_2^*)| - |I^*|$ we have

$$|V(S_2) - I| = |V(S_2^*) - I^*|$$

and $\phi$ must be bijective on $V(S_2) - I \to V(S_2^*) - I$. Thus, $\phi$ is a bijection on $V(S_1) \cup V(S_2) \to V(S_1^*) \cup V(S_2^*)$.

We are done.

□

## B Method

### B.1 Algorithm

*Proof of Lemma 2.* Since the algorithm processes subgraphs by adding one edge at a time, the theorem follows from proving that at any step in the algorithm, each subgraph in $A_i$ is disjoint and connected, then an edge can only be between two disjoint connected subgraphs or within the same connected subgraph. We prove this by induction on the number of processed edges.

*Base case*: $i = 1$. Clearly, all subgraphs consisting of a single vertex are disjoint and each such subgraph is trivially connected.

*Inductive case*: Assume true for $i \geq 1$, we want to show it is true for $i + 1$. Now at step $i + 1$, by our inductive hypothesis, all subgraphs in $A_i$ are disjoint. The next set of subgraphs $A_{i+1} = (A_i - \{S_1, S_2\}) \cup S_{1,2}$ where $S_{1,2} = S_1 \cup S_2 \cup (v_a, v_b)$, $v_a \in V(S_1)$, and $v_b \in V(S_2)$, is constructed by processing an edge $(v_a, v_b)$. Regardless of whether this edge connects two disjoint subgraphs or is within the same subgraph, in the next step, all subgraphs in $A_{i+1}$ will still be disjoint. This is because we add the new subgraph $S_{1,2}$ to form $A_{i+1}$ but remove the single subgraph (if $S_1 = S_2$) or the two subgraphs (if $S_1 \neg S_2$), to form $A_{i+1}$, that $S_{1,2}$ was connected to by the processed edge. I.e. we remove all subgraphs from $A_i$ (to form $A_{i+1}$) that the new subgraph in $A_{i+1}$ connects to. Also, since each graph $S_1$ and $S_2$ is connected, so must $S_{1,2}$ be by virtue of edge $(v_a, v_b)$.

The lemma follows. $\square$

**Remark 4.** NPA produces a sequence of encodings for a graph $G$ but when finished, set $A_{m+1}$ contains each of the largest (by inclusion) disjoint connected subgraphs of $G$. Since NPA builds encodings recursively from disjoint subgraphs, NPA constructs encodings for each such largest subgraph independently as if it is run once for each of them. Thus, proving that NPA produces iso-injective encodings for connected graphs, implies each multiset $W(G)$ and $C(G)$ is iso-injective also for disconnected graphs.

**Lemma 4.** *For any graph $S$ encoded at step $i$ on run $G$ on NPA, the function $h^j$ restricted to $V(S)$ does not change from $j = i + 1$ up to and including step $k$ (i.e. $j = k$) where $S$ is still a member of $A_k$.*

*Proof.* From the description of NPA we can tell that when a graph $S$ is encoded at step $i$ on run $G$, all $h^i$-values of $V(S)$ are updated to $h^{i+1}$-values, while all $h^{i+1}$-values of $V(G) - V(S)$ are inherited from $h^i$, and $S$ is added to $A_{i+1}$. Since all graphs in $A_k$ are disjoint (Lemma 2), the next time $h$-values of $V(S)$ will change is at step $k'$ when NPA picks $S$ from $A_{k'}$ to encode some subgraph $S_{k'} = S \cup S_2 \cup (v_a, v_b)$, updates $h^{k'+1}$-values of $V(S_{k'})$ with $V(S) \subset V(S_{k'})$, and does not include $S$ in set $A_{k'+1}$ (and never will again). On the other hand, if $S$ is not picked from $A_{k'}$ to encode $S_{k'}$ we know that $V(S_{k'}) \cap V(S) = \varnothing$ by Lemma 2 so that $h$-values of $V(S)$ do not change, i.e. $h^{k'+1}|_{V(S)} = h^{k'}|_{V(S)}$, and that $S \in A_{k'+1}$. $\square$

#### B.1.1 Proof of Theorem 8

*Proof of Theorem 8.* So we want to show that any two graphs $S_{1,2}$ run $G$ and $S_{1,2}^*$ run $G^*$ with $c(S_{1,2}) = c(S_{1,2}^*)$ are ismorphic. We prove this by double induction on the number of steps of the algorithm. This is because we need to be able to compare $c$-values that are produced at different runs of the algorithm. I.e. we want to prove a property $P(i, j)$ for all $i, j \in \mathbb{N}$, where $i$ and $j$ reflects step $i$ on first run ($G$) and step $j$ on second run ($G^*$) respectively. By the symmetry of the property, we only need to prove $P(1, 1)$ and $P(i, j) \rightarrow P(i + 1, j)$.

To be exact, the property $P(i, j)$ that we will prove consists of the following: that for any subgraph $S$ encoded at step $i' \leq i$ on run $G$ and any sugraph $S^*$ encoded at step $j' \leq j$ on run $G^*$ with $c(S) = c(S^*)$ there exists an isomorphism that

1. respects edges,

2. respects the *initial* $h^1$-values,

3. maps identical values between $h^{i'+1}(V(S))$ and $h^{j'+1}(V(S^*))$ to each other, and

4. is a bijection $V(S) \to V(S^*)$.

Since $h^1$-values are simply injective encodings of node labels, by proving this, we know the isomorphism will respect both edges and labels, and thus be a graph isomorphism.

*Base Case*: $P(0,0)$. In this case $S_{1,2}, S^*_{1,2}$ are simply vertices, and $c(S_{1,2}) = c(S^*_{1,2})$ if they have the same $h^1$-values, which means they are isomorphic in terms of $h^1$-values and edges as well as bijective. Furthermore, the isomorphism maps same values between $h^1(V(S_{1,2}))$ and $h^1(V(S^*_{1,2}))$ to each other.

*Inductive Case*: $P(i,j) \to P(i+1,j)$.

So assume we at step $i+1 > 0$ on $G$ have $S_{1,2} = S_1 \cup S_2 \cup (v_a, v_b)$, where $S_{1,2}$ is being encoded at step $i+1$.

We need to prove that for any graph $S^*_{1,2}$ encoded at step $j' \le j$ on run $G^*$ with $c(S_{1,2}) = c(S^*_{1,2})$ we have a bijective graph isomorphism between $S_{1,2}$ and $S^*_{1,2}$ that respects the edges, initial $h^1$-values, and that maps identical values between $h^{i+2}(V(S_{1,2}))$ and $h^{j'+1}(V(S^*_{1,2}))$ to each other. The reason why we only need to focus on $S_{1,2}$ is because for all other graphs encoded at step $i' < i+1$ on $G$, their $c$-values and $h^{i'+1}$-values have not changed so they are covered by our inductive hypothesis $P(i,j)$.

Now we know that $|E(S^*_{1,2})| > 0$ and $j' > 0$ because $c(S_{1,2})$ does not include the special *zero*-symbol, and therefore, neither does $c(S^*_{1,2})$. Therefore, we can also write $S^*_{1,2} = S^*_1 \cup S^*_2 \cup (v^*_a, v^*_b)$ (specifically, $(v^*_a, v^*_b)$ is the edge used to encode $S^*_{1,2}$ from the encodings of $S^*_1$ and $S^*_2$). From Lemma 2 we know $S_1, S_2, S^*_1, S^*_2$ are connected graphs.

$$c(S_{1,2}) = r(\{(c(S_1), h^{i+1}(v_a)),$$
$$(c(S_2), h^{i+1}(v_b))\},$$
$$\mathbb{1}_{S_1 = S_2})$$
$$c(S^*_{1,2}) = r(\{(c(S^*_1), h^{j'}(v^*_a)),$$
$$(c(S^*_2), h^{j'}(v^*_b))\},$$
$$\mathbb{1}_{S^*_1 = S^*_2})$$

By injectivity:

$$\big(\{(c(S_1), h^{i+1}(v_a)), (c(S_2), h^{i+1}(v_b))\},\ \mathbb{1}_{S_1=S_2}\big)$$
$$=\big(\{(c(S^*_1), h^{j'}(v^*_a)), (c(S^*_2), h^{j'}(v^*_b))\},\ \mathbb{1}_{S^*_1=S^*_2}\big)$$

and we may assume without loss of generality that

$$(c(S_1), h^{i+1}(v_a)) = (c(S^*_1), h^{j'}(v^*_a))$$
$$(c(S_2), h^{i+1}(v_b)) = (c(S^*_2), h^{j'}(v^*_b))$$

else we can just relabel the graphs.

$S_1, S_2$ are encoded before step $i+1$ on $G$ (say steps $i_1$ and $i_2$ respectively) and $S^*_1, S^*_2$ are encoded before step $j'$ on $G^*$ (say steps $j'_1$ and $j'_2$ respectively). In addition, since $S_1, S_2 \in A_{i+1}$ their $h^{i_1+1}$ and $h^{i_2+1}$ values cannot have changed before step $i+1$ (because then they would have been removed already, see Lemma 4), so $h^{i+1}|_{V(S_1)} = h^{i_1+1}|_{V(S_1)}$ and $h^{i+1}|_{V(S_2)} = h^{i_2+1}|_{V(S_2)}$ (The same holds for $S^*_1, S^*_2$). Then, we have by our inductive hypothesis two bijective isomorphisms

$$\phi_1 : S_1 \to S^*_1, \quad \phi_2 : S_2 \to S^*_2$$

with respect to edges and $h^1$-values, that maps identical values between $h^{i+1}(V(S_1))$ and $h^{j'}(V(S^*_1))$ (and between $h^{i+1}(V(S_2))$ and $h^{j'}(V(S^*_2))$) to each other, we must have

$$\forall v \in S_1, \forall v^* \in S^*_1, h^{i+1}(v) = h^{j'}(v^*) \to \phi_1(v) = v^*$$

(and similarly for $\phi_2$).

Specifically, since $h^{i+1}(v_a) = h^{j'}(v_a^*), h^{i+1}(v_b) = h^{j'}(v_b^*)$, we have

$$\phi_1(v_a) = v_a^*, \quad \phi_2(v_b) = v_b^*$$

Also we know that for all edges $(v_1, v_2) \in E(S_1), (w_1, w_2) \in E(S_2)$ we have

$$(\phi_1(v_1), \phi_1(v_2)) \in E(S_1^*), \quad (\phi_2(w_1), \phi_2(w_2)) \in E(S_2^*)$$

and the only new edge in $S_{1,2}$ is $(v_a, v_b), v_a \in V(S_1), v_b \in V(S_2)$, and the only new edge in $S_{1,2}^*$ is $(v_a^*, v_b^*), v_a^* \in V(s_1^*), v_b^* \in V(S_2^*)$.

Consider:

$$\phi(v) = \begin{cases} \phi_1(v) & \text{if } v \in V(S_1) \\ \phi_2(v) & \text{otherwise} \end{cases} \tag{4}$$

We split into two cases:

*Case 1*: $(\mathbb{1}_{S_1=S_2} = False)$. This implies that $S_1 \neq S_2$ and $S_1^* \neq S_2^*$ (where $=$ is stronger than isormorphic). By Lemma 2 we have $V(S_1) \cap V(S_2) = V(S_1^*) \cap V(S_2^*) = \varnothing$. Since $\phi$ corresponds to a graph isomorphism on the disjoint $S_1 \to S_1^*, S_2 \to S_2^*$ and the new edge is respected, $\phi$ is a graph isomorphism between $S_{1,2}$ and $S_{1,2}^*$.

In addition, since $h^{i+2}$ and $h^{j'+1}$ are injective across domains $h^{i+1}(V(S_{1,2}))$ and $h^{j'}(V(S_{1,2}^*))$ it also means that $h^{i+2}$ and $h^{j'+1}$ are injective across domains $h^{i+1}(V(S_1))$ and $h^{j'}(V(S_1^*))$. Thus, if $h^{i+2}(v) = h^{j'+1}(w)$ with $v \in V(S_1), w \in V(S_1^*)$, then $h^{i+1}(v) = h^{j'}(w)$ such that by inductive hypothesis $\phi_1(v) = w$ and thus $\phi(v) = w$ (and similarly for $S_2, S_2^*$, and $\phi_2$).

However, if there exists $v \in V(S_1), w \in V(S_2), u \in V(S_{1,2}^*)$ with $h^{i+2}(v) = h^{i+2}(w) = h^{j'+1}(u)$ we need to make sure $\phi(v) = \phi(w) = u$ (to always map identical values to each other), but then $\phi$ would not be a graph isomorphism since $v \neq w$ (we know $S_1 \cap S_2 = \varnothing$). This could also be the case for $S_1^*, S_2^*, S_{1,2}$. But by uniqueness from $r_v$ we know $h^{i+2}(V(S_1)) \cap h^{i+2}(V(S_2)) = \varnothing$ and $h^{j'+1}(V(S_1^*)) \cap h^{j'+1}(V(S_2^*)) = \varnothing$, so this cannot happen, and we can conclude that identical values across $h^{i+2}(V(S_{1,2}))$ and $h^{j'+1}(V(S_{1,2}^*))$ are always mapped to each other.

*Case 2*: $(\mathbb{1}_{S_1=S_2} = True)$. Which implies that $S_1 = S_2$ and $S_1^* = S_2^*$ (in a stronger sense than isomorphic). This means $\phi = \phi_1$. Which means that $\phi$ is bijection (no new vertices are added, only an edge), and the new edge is also respected, so $\phi$ is a graph isomorphism between $S_{1,2} \to S_{1,2}^*$ that respects $h^1$-values and edges, because $\phi_1$ does so.

In addition, $h^{i+2}$ and $h^{j'+1}$ are injective across domains $h^{i+1}(V(S_{1,2}))$ and $h^{j'}(V(S_{1,2}^*))$ with $V(S_{1,2}) = V(S_1), V(S_{1,2}^*) = V(S_1^*)$. Thus, if $h^{i+2}(v) = h^{j'+1}(w)$ with $v \in V(s_1), w \in V(S_1^*)$, then $h^{i+1}(v) = h^{j'}(w)$ such that by inductive hypothesis $\phi_1(v) = w$ and thus $\phi(v) = w$. Since $S_1 = S_2$ and $S_1^* = S_2^*$ we can conclude that identical values across $h^{i+2}(V(S_{1,2}))$ and $h^{j'+1}(V(S_{1,2}^*))$ are mapped to each other.

By Lemma 2 we know these two cases are exhaustive. Thus, $\phi$ is a bijective isomorphism between $S_{1,2}$ and $S_{1,2}^*$ with respect to edges and $h^1$-values. Furthermore, the isomorphism maps identical values across $h^{i+2}(V(S_{1,2}))$ and $h^{j'+1}(V(S_{1,2}^*))$ to each other.

Since $h^1$-values are injective with respect to node labels, we are done.

$\square$

## B.2 Existence of Required Functions

We start by proving that there exists no continuous injective function from $\mathbb{R}^2$ to $\mathbb{R}$.

**Theorem 14.** *There exists no continuous injective function $f : \mathbb{R}^2 \to \mathbb{R}$.*

*Proof.* Suppose $f : \mathbb{R}^2 \to \mathbb{R}$ is continuous. Then the image (which is an interval in $\mathbb{R}^2$) of any connected set in $\mathbb{R}^2$ under $f$ is connected. Note that this is a non-degenerate interval (a degenerate interval is any set consisting of a single real number) since the function is injective. Now, if you

remove a point from $\mathbb{R}^2$ it remains connected, but if we remove a point whose image is in the interior of the interval then the image cannot be still connected if the function is injective. □

We add some lemmas before we prove the main theorem of this section. All statements will be concerning NPA using the functions put forward in Section 3.2.

**Lemma 5.** *For NPA, all $m^1, m^2$ and h-values that appear are in $\mathbb{N}_0$*

*Proof.* We show this through an informal induction argument. Since $h_{init}(v) = l(v) \in \mathbb{N}_+$ and $c_{init}(h) = (0, 0, h + 1)$ we know that all $h^1$-values are in $\mathbb{N}_0$, and for all $c$-values created at step $0$ we have $m^1, m^2 \in \mathbb{N}_0$. Now since new $m$-values are created from $m_1^1, m_1^2, m_2^1, m_2^2 \in \mathbb{N}_0$ through $m_{1,2}^1 = m_1^2 + m_2^2 + 1 \in \mathbb{N}_0, m_{1,2}^2 = 2(m_1^2 + m_2^2 + 1) \in \mathbb{N}_0$ it is not hard to see that all $m^1, m^2$ that appear will be in $\mathbb{N}_0$. Similarly, new $h^{i+1}$-values are created from $h^i$-values through $h^{i+1}(v) = h^i(v) \in \mathbb{N}_0$ or $h^{i+1}(v) = h^i(v) + m^1 \in \mathbb{N}_0$ (since $m^1 \in \mathbb{N}_0$), so all $h$-values will be in $\mathbb{N}_0$. □

**Lemma 6.** *For any graph $s_k$ encoded by Algorithm 2 at step $i$ on run $G$ we have $m_k^2 > \max(h^{i+1}(V(S_k)))$ and each value in $h^{i+1}(V(S_k)) = r_v(c(S_k), h^i(V(S_k)))$ is unique.*

*Proof.* We will prove this by strong induction on the number of steps $i$ of the algorithm on run $G$. Property $P(i)$ is that any graph $S_k$ encoded at step $i$ on run $G$:

- $m_k^2 > \max(h^{i+1}(V(S_k)))$, and

- each value in $h^{i+1}(V(S_k)) = r_v(c(S_k), h^i(V(S_k)))$ is unique

*Base Case*: $P(0)$. This means $S_k$ consists of a single vertex $v$. Thus, $h^1(V(S_k)) = \{l(v)\} \subset \mathbb{N}_+$ and it is unique. Consequently, $m_k^2 = l(v) + 1 > 0$, such that $m_k^2 > \max(h^1(V(S_k)) = l(v)$. We also note that $m_k^1 = 0$.

*Inductive Case*: $(\forall i' \leq i, P(i')) \rightarrow P(i+1)$.

Since $i + 1 > 0$ we have $|E(s_k)| > 0$ so we can write $V(S_{1,2}) := V(S_k) = V(S_1) \cup V(S_2)$, where $S_1, S_2$ were encoded before step $i + 1$, say step $i_1$ and $i_2$ respectively. By inductive hypothesis, this means that all values in $h^{i_1+1}(V(S_1))$ and all values in $h^{i_2+1}(V(S_2))$ are unique, and since $S_1, S_2 \in A_{i+1}$, by Lemma 4, these $h$-values cannot have changed before step $i + 1$ (i.e. $h^{i_1+1}|_{V(S_1)} = h^{i+1}|_{V(S_1)}, h^{i_2+1}|_{V(S_2)} = h^{i+1}|_{V(S_2)}$). Thus, each value in $h^{i+1}(V(S_1))$ and each value in $h^{i+1}(V(S_2))$ is unique. By injective hypothesis we also know that

$$m_1^2 > \max(h^{i+1}(V(S_1))), \; m_2^2 > \max(h^{i+1}(V(S_2)))$$

From Lemma 5, we know $m_1^2, m_2^2 \in \mathbb{N}_0$ and all $h$-values in $\mathbb{N}_0$, i.e. they are non-negative.

Now we have, with $m_{1,2}^1 = m_1^2 + m_2^2 + 1 > 0$, that

$$h^{i+2}(V(S_{1,2})) := r_v(c(S_{1,2}), h^{i+1}(v)) =$$
$$\left\{ \begin{array}{ll} h^{i+1}(v) + m_{1,2}^1, & \text{if } v \in V(S_1) \\ h^{i+1}(v), & \text{else} \end{array} \right\}$$

This means now that each value in $h^{i+2}(V(S_1))$ and each value in $h^{i+2}(V(S_2))$ is unique. This is easier to see for $h^{i+2}(V(S_1))$ because $r_v$ is an injective function on the values of $h^{i+1}(V(S_1))$ which we know are all unique. However, since

$$m_{1,2}^1 > \max(h^{i+1}(V(S_2))), \; \min(h^{i+1}(V(S_1))) \geq 0$$

$r_v$ is also injective on $h^{i+1}(V(S_2))$. To prove this, suppose $r_v(h^{i+1}(v)) = r_v(h^{i+1}(w))$ with $v, w \in V(S_2)$, then $h^{i+1}(v) = h^{i+1}(w)$ unless, w.l.o.g, $v \in V(S_1), w \notin V(S_1)$ from which we reach a contradiction since $\min(h^{i+1}(V(S_1))) + m_{1,2}^1 > \max(h^{i+1}(V(S_2)))$.

Since $\max(h^{i+1}(V(S_1))) + m_2^2 + 1 > \max(h^{i+1}(V(S_2)))$ we have
$$\max(h^{i+2}(V(S_{1,2}))) = \max(h^{i+1}(V(S_1)))$$
$$+ m_1^2 + m_2^2 + 1$$
$$< 2m_1^2 + m_2^2 + 1$$

Since $m_{1,2}^2 = 2m_1^2 + 2m_2^2 + 2 > 0$ this means that $\max(h^{i+2}(V(S_{1,2}))) < m_{1,2}^2$. We can also conclude $m_{1,2}^1, m_{1,2}^2 \in \mathbb{N}_+$.

By Lemma 2, we know that either $S_1 = S_2$ or $S_1 \cap S_2 = \varnothing$. If $S_1 = S_2$, then $V(S_{1,2}) = V(S_1) = V(S_2)$ such that $h^{i+2}|_{V(S_{1,2})} = h^{i+1}|_{V(S_1)} + m_{1,2}^1$, which means that each value in $h^{i+2}(V(S_{1,2}))$ is unique because each value in $h^{i+1}(V(S_1))$ is unique. Thus we are done, and we now assume that $S_1 \cap S_2 = \varnothing$.

This means that $V(S_1) \cap V(S_2) = \varnothing$ and
$$h^{i+2}(V(S_1)) \cap h^{i+2}(V(S_2)) = \varnothing$$
since $m_{1,2}^1 > m_1^2 + m_2^2 > \max(h^{i+1}(V(S_2)))$, $\max(h^{i+2}(V(S_2))) = \max(h^{i+1}(V(S_2)))$. Thus, all values in
$$h^{i+2}(V(S_{1,2})) = h^{i+2}(V(S_1)) \sqcup h^{i+2}(V(S_2))$$
are unique.

Thus we have proved $P(i+1)$. $\qquad\square$

**Corollary 4.** *This also means that $m_k^1 = 0$ if and only if $|E(S_k)| = 0$ (i.e. in the base case). Thus, it serves as the required $zero$-symbol.*

Armed with this lemma we will now prove the following:

**Lemma 7.** *For all graphs $S, S^*$ encoded at step $i$ run $G$ and $j$ run $G^*$ respectively with $c := c(S) = c(S^*)$, $r_v(c, \cdot)$ is injective across domains $h^i(V(S))$ and $h^j(V(S^*))$.*

**Remark 5.** We reiterate, with a function $f : X \to Y$ being injective across domain $X_1$ and $X_2$ with $X_1, X_2 \subset X$, we mean that for all $x_1 \in X_1, x_2 \in X_2$ with $f(x_1) = f(x_2)$ we have $x_1 = x_2$.

*Proof.* First if $i = 0$ or $j = 0$ we know that both $i = j = 0$ due to the $zero$-symbol, and then it is vacuously true, because $h^0$ does not exist and $r_v$ is not applied. So we assume $i, j > 0$.

Since $i, j > 0$ we have $V(S) = V(S_1) \cup V(S_2)$, $V(S^*) = V(S_1^*) \cup V(S_2^*)$. We also know $(m^1, m^2) = (m_*^1, m_*^2)$. By Lemma 2 we know that either $S_1 = S_2$ or $S_1 \cap S_2 = \varnothing$.

If $S_1 = S_2$, then since $c(S) = c(S^*)$ we also have $S_1^* = S_2^*$, which means that $V(S) = V(S_1) = V(S_2)$ and $V(S^*) = V(S_1^*) = V(S_2^*)$. This means that $r_v(c, h) = h + m^1 = h + m_*^1$, which then is injective and in particular injective across $h^i(V(S))$ and $h^j(V(S^*))$. Thus, we now assume that $S_1 \cap S_2 = \varnothing$.

This means that $V(S_1) \cap V(S_2) = \varnothing$. Now suppose
$$r_v(c, h_a^i) = r_v(c, h_b^j)$$
with $h_a^i \in h^i(V(S)), h_b^j \in h^j(V(S^*))$. Consider two cases:

*Case 1*: $h_a^i \in h^i(V(S_1))$. Then
$$r_v(c, h_a^i) = h_a^i + m^1 = h_a^i + m_*^1$$
Since $m_*^1 > \max(h^j(V(S_2^*))) \geq 0$ and $h_a^i \geq 0$ (Lemma 6 and 5) we must have $h_b^j \in h^j(V(S_1^*))$ such that
$$r_v(c, h_b^j) = h_b^j + m_*^1$$
Because else
$$r_v(c, h_b^j) = h_b^j < m_*^1 < r_v(c, h_a^i)$$
This implies that $h_a^i = h_b^j$.

*Case 2*: $h_a^i \notin h^i(V(S_1))$ which means that $h_a^i \in h^i(V(S_2))$. Suppose by contradiction that $h_b^j \in h^j(V(S_1^*))$ then
$$r_v(c, h_a^i) = h_a^i = r_v(c, h_b^j) = h_b^j + m_*^1 = h_b^j + m^1$$
But since $m^1 > \max(h^i(V(S_2)) \geq 0$ and $h_b^j \geq 0$ (Lemma 6 and 5) we get a contradiction. This means $h_b^j \notin h^j(V(S_1^*)), h_b^j \in h^j(V(S_2^*))$ such that
$$r_v(c, h_a^i) = h_a^i = r_v(c, h_b^j) = h_b^j$$

We are done. $\qquad\square$

Consider the following functions:

$$\tau(i,j) = \frac{(i+j)(i+j+1)}{2} + j, \quad \rho(i,j) = (i+j, ij)$$

**Lemma 8.** *Two claims:*

- $\tau : \mathbb{R} \times \mathbb{R} \to \mathbb{R}$ *is continuous and injective in* $\mathbb{N} \times \mathbb{N} \to \mathbb{N}$.

- $\rho : \mathbb{R} \times \mathbb{R} \to \mathbb{R}$ *is continuous and injective in* $\{\{i,j\} \mid i,j \in \mathbb{N}\} \to \mathbb{N}^2$.

*Proof.* $\tau$ is the well-known Cantor Pairing Function, see for example Wikipedia for proof of its bijective properties on $\mathbb{N}^2 \to \mathbb{N}$, it is clearly continuous on $\mathbb{R}^2 \to \mathbb{R}$.

$\rho$ is cleary continuous in $\mathbb{R}^2 \to \mathbb{R}^2$ and if $i,j \in \mathbb{N}$ then $\rho(i,j) \in \mathbb{N}^2$. We will prove that it is injective in $\{\{i,j\} \mid i,j \in \mathbb{N}\} \to \mathbb{N}^2$:

Suppose $(i+j, ij) = (x,y)$ we want to express $i$ and $j$ in terms of $x$ and $y$. Rearranging and substituting, we get $i = x - j \Rightarrow (x-j)j = y \Rightarrow j^2 - xj + y = 0$. Using the quadratic formula, and by symmetry, we get

$$j = \frac{x \pm \sqrt{x^2 - 4y}}{2}, \quad i = \frac{x \pm \sqrt{x^2 - 4y}}{2}$$

If $j = \frac{x+\sqrt{x^2-4y}}{2}, i = \frac{x-\sqrt{x^2-4y}}{2}$ (or other way around) the conditons $i+j = x, ij = y$ holds. But if $j = \frac{x+\sqrt{x^2-4y}}{2} = i = \frac{x+\sqrt{x^2-4y}}{2}$ then $i+j = x + \sqrt{x^2-4y}$ and $ij = \frac{x^2}{4} + x\sqrt{x^2-4y} + \frac{x^2-4y}{4}$ and conditions hold iff $x^2 = 4y$ which takes us back to our previous case. Similarly, if $j = \frac{x-\sqrt{x^2-4y}}{2} = i = \frac{x-\sqrt{x^2-4y}}{2}$ then $i+j = x - \sqrt{x^2-4y}, ij = \frac{x^2}{4} - x\sqrt{x^2-4y} - \frac{x^2-4y}{4}$ and conditions hold iff $x^2 = 4y$ which again takes us back to our first case. Thus, we have proved that $\rho$ is injective. $\square$

**Lemma 9.** *In the above setup, there exists a continuous and bounded function* $r : \mathbb{R}^9 \to \mathbb{R}$ *that is injective in* $\{\mathbb{N}^4, \mathbb{N}^4\} \times \mathbb{N}$. *Namely,*

$$r(y_1, h_1, m_1, n_1, y_2, h_2, m_2, n_2, b) = \tau\big(\tau\big(\rho(\tau^4(y_1, h_1, m_1, n_1), \tau^4(y_2, h_2, m_2, n_2))\big), b\big)$$

*Proof.* The proof follows from Lemma 8. $\square$

**Lemma 10.** *For the functions defined in Section 3.2 and in this section, when used in NPA, we always have (i)* $h^j(v) \in \mathbb{N}_0$ *and (ii)* $c(S_k) = (y_k, m_k^1, m_k^2) \in \mathbb{N}_0 \times \mathbb{N}_0 \times \mathbb{N}_0 = \mathbb{N}_0^3$.

*Proof.* (i) $h^j(v) \in \mathbb{N}_0$ follows immediately from Lemma 5. Note that (ii) is true for all $c$-values encoded at step 0 in NPA via $c_{init}$ since all $h$-values are in $\mathbb{N}_0$, also we know that all $m_k^1, m_k^2 \in \mathbb{N}_0$ from Lemma 5. Thus, the only thing we need to consider is the subsequent application of $r$, and it is applied to $h$-values, $c$-values, and $\{0,1\}$-indicators, all of which are in $\mathbb{N}_0$, to create new $c$-values. Since $r$ takes $(\mathbb{N}_0)^*$ to $(\mathbb{N}_0)^*$, which can be seen by inspection, the lemma follows. $\square$

**Lemma 11.** *The* $r_c$ *function with the* $r$*-function from Lemma 9 is injective in all its variables.*

*Proof.* Suppose

$$r_c(\{(c_1^1, h_1^1), (c_2^1, h_2^1)\}, b_1) = r_c(\{(c_1^2, h_1^2), (c_2^2, h_2^2)\}, b_2)$$

Where

$$c_1^1 = (y_1^1, m_1^1, n_1^1), \ c_2^1 = (y_2^1, m_2^1, n_2^1)$$
$$c_1^2 = (y_1^2, m_1^2, n_1^2), \ c_2^2 = (y_2^2, m_2^2, n_2^2)$$

This means that

$$\big(r(y_1^1, h_1^1, m_1^1, n_1^1, y_2^1, h_2^1, m_2^1, n_2^1, b_1),$$
$$n_1^1 + n_2^1 + 1,\ 2n_1^1 + 2n_1^1 + 2\big) =$$
$$\big(r(y_1^2, h_1^2, m_1^2, n_1^2, y_2^2, h_2^2, m_2^2, n_2^2, b_2),$$
$$n_1^2 + n_2^2 + 1,\ 2n_1^2 + 2n_2^2 + 2\big)$$

Thus, from Lemma 9 we know $r$ is injective in $\{\mathbb{N}^4, \mathbb{N}^4\} \times \mathbb{N}$. By Lemma 10 we know all input to $r$ are in $\mathbb{N}_0$, thus, $r$ is injective, which gives us

$$\big(\{(c_1^1, h_1^1), (c_2^1, h_2^1)\},\ b_1\big) = \big(\{(c_1^2, h_1^2), (c_2^2, h_2^2)\},\ b_2\big)$$

and we are done. $\qquad\square$

**Lemma 12.** *For Algorithm 2 there exists functions $r_v$, $r_c$, $h_{init}, c_{init}$ that satisfies the requirements put forward in Theorem 8.*

*Proof.* Consider the functions defined in Section 3.2 and in this section, as well as the results. The lemma follows. $\qquad\square$

## B.3 Corollaries

We add a remark about the subgraphs that are encoded during runs of NPA on a graph $G$.

**Remark 6.** On one run of NPA on graph $G$, the multiset $W(G)$ encodes a collection of subgraphs of $G$, for example, these subgraphs always include the vertices and the largest (by inclusion) connected subgraphs. The order in which edges are processed determines which other subgraphs that are encoded, but it is not too hard to see that if NPA is run on all possible orders on edges, and without NPA changing the order, it will encode each combination of disjoint connected subgraphs. Since any subraph consists of a collection of disjoint connected subgraphs, it will indirectly encode all possible subgraphs.

Full proof of Lemma 3

*Proof.* (From [25]). We first prove that there exists a mapping $f$ so that $\sum_{x \in X} f(x)$ is unique for each multiset $X$ bounded size. Because $\mathcal{X}$ is countable, there exists a mapping $Z : \mathcal{X} \to \mathbb{N}$ from $x \in \mathcal{X}$ to natural numbers. Because the cardinality of multisets $X$ is bounded, there exists a number $N \in \mathbb{N}$ so that $|X| < N$ for all $X$. Then an example of such $f$ is $f(x) = N^{-Z(x)}$. This $f$ can be viewed as a more compressed form of an one-hot vector or $N$-digit presentation. Thus, $h(X) = \sum_{x \in X} f(x)$ is an injective function of multisets. $\phi(\sum_{x \in X} f(x))$ is permutation invariant so it is a well-defined multiset function. For any multiset function $g$, we can construct such $\phi$ by letting $\phi(\sum_{x \in X} f(x)) = g(X)$. Note that such $\phi$ is well-defined because $h(X) = \sum_{x \in X} f(x)$ is injective. $\qquad\square$

**Corollary 5.** *There exists a function $f$ such that any two graphs $G$ and $H$ in $\mathcal{G}_b$ are isomorphic if $\sum_{w \in W(G)} f(w) = \sum_{w \in W(H)} f(w)$.*

**Remark 7.** Given a graph isomorphism class $[S]$ and assuming NPA does not change the order of the edges, there is a Turing-decidable function $f_{[S]} : \mathcal{G} \to [0, 1]$ that on input $G$ returns 1 if there exists $S \in [S], H \in [G]$ with $S \subset H$ and 0 otherwise; in pseudo-code:

$$f_{[S]} \text{ on input } G,$$
$$\forall H \in [G], \forall S \in [S],$$
$$\text{if } W(S) \subset W(H) \text{ return } 1,$$
$$\text{return } 0$$

which is Turing-decidable since for any $G \in \mathcal{G}$ all such sets $[G], [S], W(H), W(S)$ are finite. However, a similar function for detecting the presence of a subgraph in isomorphism class $[S]$

in graph $G$ given we only have one encoding $E(G)$ for all of $G$ must not exist. Without some subset-information in the encoding we are left to (pseudo-code):

$$f_{[S]} \text{ on input } G,$$
$$\forall H \in \mathcal{G}, \exists S \in [S], S \subset H,$$
$$\text{if } E(G) = E(H) \text{ return } 1,$$
$$\text{return } 0$$

which is Turing-recognizable but not Turing-decidable, because the number of graphs $H \in \mathcal{G}$ that contain subgraphs in $[S]$ is infinite. This points to the strength of having the encoding of a graph $G$ coupled with encodings of its subgraphs.

## B.4 Use of Neural Networks

We make use of the following functions:

$$c_{init}(i) = (0, 0, i + 1)$$
$$f_1(i, j) = i + j + 1$$
$$f_2(i, j) = 2i + 2j + 2$$
$$r(y_1, h_1, m_1, n_1, y_2, h_2, m_2, n_2, b) =$$
$$\tau\big(\tau\big(\rho(\tau^4(y_1, h_1, m_1, n_1), \tau^4(y_2, h_2, m_2, n_2))\big), b\big)$$
$$r_v(\dots, m, h, \mathbb{1}_{ind}) = h + \mathbb{1}_{ind}m$$

Where

$$\tau(i, j) = \frac{(i + j)(i + j + 1)}{2} + j, \quad \rho(i, j) = (i + j, ij)$$

To a lesser extent we use

$$f_3(i) = N^{-i}$$

By Theorem 3, NNs can perfectly approximate any function on a finite domain so the case of $\mathcal{G}_b$ is straightforward. However, for countably infinite $\mathcal{G}$ the situation is different. Note that these functions are continuous (in $\mathbb{R}^*$) but not bounded and that we are applying these functions recursively and would want both the domain and the image to be bounded iteratively. Without losing any required properties we can compose these functions, $f$, with an injective, bounded, and continuous function with continuous inverse such as Sigmoid, $\sigma$, in the following way $f^* = \sigma \circ f \circ \sigma^{-1}$, and use $h_{init}(l(v)) = \sigma(l(v))$. Then these functions can be pointwise approximated by NNs.

**Lemma 13.** $\sigma : \mathbb{R} \to (0, 1)$, $\sigma(x) = \frac{1}{1+e^x}$ is continuous, bounded, and injective. Also, its inverse $\sigma^{-1} : (0, 1) \to \mathbb{R}$ is continuous and injective.

*Proof.* $\sigma$ is continuous since the exponential function is continuous, and it is clearly bounded with $\operatorname{im}(\sigma) = (0, 1)$. Furthermore, its inverse is $\sigma^{-1}(x) = \ln(\frac{1-x}{x}) : (0, 1) \to \mathbb{R}$, thus it is injective. Since ln is continuous so is $\sigma^{-1}$, and since $\sigma^{-1}$ is the inverse of a function, it is injective. $\square$

The required functions then become:

$$c_{init}^* : (0, 1) \to (0, 1), \; c_{init}^* = \sigma \circ c_{init} \circ \sigma^{-1}$$
$$f_1^* : (0, 1)^2 \to (0, 1), \; f_1^* = \sigma \circ f_1 \circ \sigma^{-1}$$
$$f_2^* : (0, 1)^2 \to (0, 1), \; f_2^* = \sigma \circ f_2 \circ \sigma^{-1}$$
$$r^* : \{(0, 1)^4, (0, 1)^4\} \times (0, 1) \to (0, 1), \; r^* = \sigma \circ r \circ \sigma^{-1}$$
$$r_v^* : (0, 1)^3 \to (0, 1), \; r_v^* = \sigma \circ r_v \circ \sigma^{-1}$$

It follows from the setup and Lemma 10 that if $\operatorname{im}(h_{init}) \subset \{\sigma(i) \mid i \in \mathbb{N}\}$ then all these functions maintain their required properties. All these functions are continuous and bounded (iteratively on $(0, 1)$ by $(0, 1)$) in $\mathbb{R}^*$. Thus, by Theorem 6, they can be pointwise approximated by a NN. Yet,

Table 3: Edge-orders and levels.

| Datasets: | | NCI1 | MUTAG | PROTEINS | PTC |
|---|---|---|---|---|---|
| Avg # nodes: | | 30 | 18 | 39 | 26 |
| Avg # edges: | | 32 | 20 | 74 | 26 |
| $O$(median # edge-orders): | degs-and-labels | $10^7$ | $10^5$ | $10^{13}$ | $10^5$ |
| $O$(median # edge-orders): | two-degs | $10^9$ | $10^7$ | $10^{23}$ | $10^6$ |
| $O$(median # edge-orders): | one-deg | $10^{20}$ | $10^{14}$ | $10^{36}$ | $10^{16}$ |
| $O$(median # edge-orders): | none | $10^{31}$ | $10^{17}$ | $10^{62}$ | $10^{23}$ |
| Avg samples # levels: | degs-and-labels | 12 | 11 | 41 | 9 |
| Avg samples # levels: | two-degs | 12 | 10 | 41 | 9 |
| Avg samples # levels: | one-deg | 14 | 11 | 41 | 13 |
| Avg samples # levels: | none | 12 | 14 | 39 | 13 |

for $f_3$ the situation is a little different because we care about the sum $\sum_{x \in X} f_3(x)$ over a bounded multiset $X$. However, note that all the domain consists of $\mathbb{N}_0$ so $f_3$ is bounded by $(0,1]$. Thus we can pointwise approximate

$$f_3^* : (0,1) \to (0,1] : f_3^* = f_3 \circ \sigma^{-1}$$

which suffices, and if $X$ is bounded, so is the sum.

However, it also follows, due to the use of $\sigma$, that the pointwise approximation error is going to be more likely to cause problems for large values.

### B.4.1 Approximation Error and its Accumulation

Recursive application of a NN might increase the approximation error. We have the following equations describing successive compositions of a NN $\varphi$:

$$\begin{aligned}
&\|f(f(x)) - \varphi(\varphi(x))\| \\
&= \|f(f(x)) - \varphi(f(x) + \epsilon)\| \\
&= \|f(f(x)) - f(f(x) + \epsilon) + \epsilon\|
\end{aligned}$$

Future work should investigate the effects of this likely accumulation.

### B.5 Class-Redundancy, Sorting, Parallelize, and Subgraph Dropout

Again, the class-redundancy in the algorithm and functions we propose enters at the sort functions $s_e$ (sorts edges) and $s_v$ (sorts nodes within edges). Thus, a loose upper bound on the *class-redundancy* is $O((m!)2^m)$. However, a more exact upper bound is $O((t_1!)(t_2!)\ldots(t_k!)(2^p))$, where $t_i$ are the sizes of the consecutive ties for the sorted edges, and $p$ (bounded by $m$) is the number of ties for the sorting of nodes within edges. An even better upper bound is

$$O((t_{1,1}!)\ldots(t_{1,l_1}!)(t_{2,1}!)\ldots(t_{k,l_k}!)(2^p))$$

where each $t_{i,j}$ is the number of ties within group $j$ of groups of subgraphs that could be connected within the tie $i$. The order in between disconnected tied subgraph groups does not affect the output.

In Table 3 you can find #edge-orders, that is $O((t_{1,1}!)\ldots(t_{1,l_1}!)(t_{2,1}!)\ldots(t_{k,l_k}!))$, and #levels on some datasets.

### B.6 Neural Networks

For NPBA we let $c(S_i) = (c_i^1, c_i^2)$ be the encoding for a subgraph $S_i$ and use for $r_c$:

$$i = \sigma(W_i(c_0^2 + c_1^2) + b_i)$$
$$f_1 = \sigma(W_f c_0^2 + b_f)$$
$$f_2 = \sigma(W_f c_1^2 + b_f)$$
$$g = \tanh(W_g(c_0^2 + c_1^2) + b_g)$$
$$o = \sigma(W_o(c_0^2 + c_1^2) + b_o)$$
$$c_{1,2}^1 = f_1 * c_0^1 + f_2 * c_1^1 + i * g$$
$$c_{1,2}^2 = o * \tanh(c_{1,2}^1)$$

For the NPA we use for $r_c(\{(c(S_1), h_1), (c(S_2), h_2)\}, s := \mathbb{1}_{S_1=S_2})$:

$$i = \sigma(W_{i,h}(h_1 + h_2) + W_{i,c}(c_1^2 + c_2^2) + W_{i,s}s + b_i)$$
$$f_1 = \sigma(W_{f,h}h_1 + W_{f,c}c_1^2 + W_{f,s}s + b_f)$$
$$f_2 = \sigma(W_{f,h}h_2 + W_{f,c}c_2^2 + W_{f,s}s + b_f)$$
$$g = \tanh(W_{g,h}(h_1 + h_2) + W_{g,c}(c_1^2 + c_2^2) + W_{g,s}s + b_g)$$
$$o = \sigma(W_{o,h}(h_1 + h_2) + W_{o,c}(c_1^2 + c_2^2) + W_{o,s}s + b_o)$$
$$c_{1,2}^1 = f_1 * c_1^1 + f_2 * c_2^1 + i * g$$
$$c_{1,2}^2 = o * \tanh(c_{1,2}^1)$$

Where $s = \mathbb{1}_{S_1=S_2}$ and the encoding for a subgraph $S_i$ is $c(S_i) = (c_i^1, c_i^2)$ and the $h$-value of a node $v_j$ is encoded by $h_j$ (so $h_1$ and $h_2$ above encode $h(v_a)$ and $h(v_b)$ respectively).

For $r_v(c(S_{1,2}), h_v, t := \mathbb{1}_{v \in V(s_1)})$ we use (with a different set of weights)

$$i = \sigma(W_{i,c}c_{1,2}^2 + W_{i,t}t + b_i)$$
$$f = \sigma(W_{f,c}c_{1,2}^2 + W_{f,t}t + b_f)$$
$$g = \tanh(W_{g,c}c_{1,2}^2 + W_{g,t}t + b_g)$$
$$o = \sigma(W_{o,c}c_{1,2}^2 + W_{o,t}t + b_o)$$
$$h_v = f * h_v + i * g$$

Where $t = \mathbb{1}_{v \in V(s_1)}$. Intuitively, we make it easy for the label to flow through.

## C  Experiments

### C.1  Synthetic Graphs

The ordering of the nodes of a graph $G$ are randomly shuffled before $G$ is feed to NPA and the output depends to some extent on this order. This makes it hard for a NN to overfit to the features that NPA produces on a training set. For datasets where the class-redundancy is large (e.g regular graphs) NPA might never produce the same encoding between the gradient steps and the training accuracy evaluation. This may cause NNs to overfit to the encodings NPA produces during the batch updates and underfit the encodings produced for evaluation of training accuracy. Even during training, NPA (and NPBA) might never produce the same representation for the same graph twice.

### C.2  Experiment Details

We try and compare algorithms at the task of classifying graphs. Every dataset maps each of its graphs to a ground-truth class out of two possible classes.

We report the average and standard deviation of validation accuracies across the 10 folds within the cross-validation. We use the Adam optimizer with initial learning rate 0.01 and decay the learning

rate by 0.5 every 50 epochs. We tune the number of epochs as a hyper-parameter, i.e., a single epoch with the best cross-validation accuracy averaged over the 10 folds was selected.

In the experiments, the $W(G)$ features are summed and passed to a classify-NN consisting of either one fully-connected layer and a readout layer (for MUTAG, PTC, and PROTEINS) or two fully-connected layers and a readout layer (for NCI1), where the hidden-dim of the fully connected layers is of size $d_{hidden}$. For $h_{init}$ we use a linear-layer followed by a batchnorm (for MUTAG, PTC, and PROTINES) or a linear-layer followed by activation function and batchnorm (for NCI1). In addition, for NCI1 we used dropout=0.2 after each layer in the classify-net and on the vectors of $W(G)$ before summing them.

Also, in our experiments we skipped including the $w_i$ features for the single nodes. In fact, all datasets consist of connected graphs.

For the NPBA tree-lstm the dimensions of $c^1$ and $c^2$ is $d_{hidden}$. For the NPA the dimensions of $c^1$ and $c^2$ is $d_{hidden}$ and the dimension of $h$ is $d_{hidden}/2$.

We used the following settings for $d_{hidden}$ and batch size:

- PTC, PROTEINS, and MUTAG we used $d_{hidden} = 16$, and batch-size=32.
- NCI1 we used $d_{hidden}$ = 64, and batch-size=128.