[Reviews · NeurIPS 2020]

Review 1

Summary and Contributions: The paper studies the problem of discriminating graph isomorphism classes. More specifically, the paper shows that any real function on bounded graphs (graphs of bounded size) can be approximated by an injective function that is computable by a NN. For unbounded graphs, the paper gives a weaker approximation under certain restrictions. The paper also presents an algorithm for approximating function on bounded graphs. The paper validates the approach on benchmark datasets in graph classifications, and compares the proposed approach to existing approaches.

Strengths: This is a solid work all around, studying a very important problem that is relevant to machine learning. The paper presents a solid theoretical framework, implement their algorithm, and validate it and compare it to existing results.

Weaknesses: The paper is not written well. The writing of certain sections is very terse and they are difficult to read.

Correctness: I believe so.

Clarity: No, the writing can be significantly improved. The problem can be motivated better by starting with some examples. The introduction is very short and lacks good examples; instead, it describes the problem at a high level. The preliminaries section is very dense, and contains theorems (e.g. thm. 1) that are not referenced (and it is not clear if they are part of the contribution of the paper).

Relation to Prior Work: Yes. The paper discusses existing approaches, and the empirical results include comparisons to previous works.

Reproducibility: No

Additional Feedback: I have read the authors' rebuttal.


Review 2

Summary and Contributions: The paper develops graph representations with injective properties to construct universal approximations, which are then used for graph classification tasks

Strengths: This has been a very active research topic, and the main ideas of the paper are quite interesting. The technical contribution is strong

Weaknesses: The paper is very poorly written and difficult to follow. A lot of notions are not properly explained. The experimental results are not particularly significant, relative to past work

Correctness: The presentation makes it hard to verify correctness. The experimental section does not seem very strong (relative to past work), and is not presented very well.

Clarity: Very poor

Relation to Prior Work: Pretty good. A recent paper by Loukas (What graph neural networks cannot learn: depth vs width), ICLR 2020 is also very relevant, and should be discussed. Other work, such as [14] is not discussed adequately

Reproducibility: Yes

Additional Feedback: page 1, last para: this is confusing to read. The reference cited here is Babai's paper. Why is the slowness of current graph isomorphism algorithms relevant to the problem of producing isomorphism-injective graph representations? Definition 3: a minor point, but it is useful to say what is meant by size (e.g., #edges, or size of description of the graph) After definition 7, it is useful to formally define the notion of "universal function approximator", as this could be interpreted in different ways The notation of multi-function in Definition 6 uses a double arrow, but it doesn't seem to get used consistently like that. It is used with a single arrow in Definition 7. Also the notion is confusing, since it is not a function into the range but into its power set. Page 3: the comment that the running time in algorithm is a function of the input seems odd. The run time of any algorithm will be in terms of the input Section 2.3 is written quite poorly. The discussion about prior results (Theorem 5), and how the new results in the paper fit in, needs to be improved page 4: The justification for Postulate 1 is not very clear. Why is the detection of shared subgraphs relevant? Are these induced or non-induced subgraphs? Finally, this needs to be connected to the isomorphism problem. The authors should give references for the canonization algorithm mentioned. Also, the #subgraphs can be larger if subgraphs of different sizes are considered Algorithm 1: what is d_c? And what does an extended function mean? Theorem 7: "run" should be input graph. The notion of subgraph A\in G is not very precise. Is it induced or non-induced subgraphs? Algorithm 2 is very poorly written, with lots of notation, e.g., s_e(.), s_v, etc, which is not defined. There is no intuitive description given for helping the reader understand it The experiments section has very little detail, and the supplementary section also does not give too much information. A reader has to look at prior papers to really under stand what these datasets are, what the labels are, and what the prediction task is Are the improvements in the experiments sufficiently better than the results of [24] in Table 1? For two of the four datasets, the performance is the same. Not sure how this will generalize ---------------------------- I have read the author response, and some of the concerns have been addressed.


Review 3

Summary and Contributions: The paper shows that iso-injective functions combined with reLU neural nets allow for universal function approximation on graphs, i.e. to approximate arbitrary functions from certain sets of graph isomorphism classes to the reals. They provide a practical algorithm within their framework and show that it achieves results that are comparable to recent works, while theoretically being an universal function approximator.

Strengths: The theoretical framework is novel and elegant. Furthermore, the connection between iso-injective functions on graphs and multivalued functions on graph isomorphism classes can be useful for the community to develop novel methods on these grounds. The practical algorithm seems to perform reasonably well.

Weaknesses: The paper is difficult to read at times, as some connections between results are not mentioned. The ground definitions need to be rewritten to make Lemma 1 (the basis of the overall claim of the paper) true, as the authors mention in their rebuttal, this can be easily achieved. Algorithm 2 needs to be analyzed in detail for runtime. In particular, Corr. 1 suggests that Alg. 2 has superpolynomial runtime, that there is a second paper hidden here (in case you are faster than Babai), or that the claim of Corr. 1 is incorrect. Furthermore, the connection between sections 2.1, 2.2 and 2.3, 2.4 is not clear.

Correctness: As mentioned above, I have doubts about Corr. 1, without being able to give a counterexample, due to time constraints. Furthermore, I am not able to follow the argumentation between Theorem 5 and Remark 1. My concerns regarding Definition 1 and Lemma 1 have been addressed in the rebuttal of the authors. ( These were: You define graphs as labeled with function $l: V(G) \to \mathbb{N}$. Without restricting this adequately, I doubt that Lemma 1 is correct, i.e., I assume that the set of all labeled graphs on at most n vertices is infinite, as I can easily label a singleton graph (i.e., that contains a single vertex) with any natural number. Hence, I have infinitely many graphs in $\mathcal{G}_1$ and $\bold\mathcal{G}_1$, although 'each graph is finite in terms of nodes, edges, and labels'. The same thing, by the way, applies also to the vertex sets of your graphs, if you don't restrict the vertex in $\mathcal{G}_1$ to be 1. I think that this can be fixed by restricting the labels to the set $[b] = \[1, ..., b\}$ \[ \mathcal{G}_b = \{ G=(V(G), E(G), l) | V(G) \subseteq [b] \wedge l: V(G) \to [b] \} \] (note by the way, that size of G is not defined in your paper and I'll assume number of nodes to be what you mean) ) As the authors certify in the rebuttal, my concern below was correct, but only the weaker result is what they have intended to show. However, I am still at a loss how Remark 1 results from the argumentation below Thm. 5. ( Thm 5 / Remark 1: I am not sure how Remark 1 results from the argumentation below Thm. 5. Furthermore, it seems unclear to me why the point $p$ to which the identified subsequence converges (called $p=Alg([G]*)$ ) would actually be in img(Alg), i.e., does there exist a graph $H$ such that p = Alg([H]) ? ) Hence I cannot certify the correctness of the following claims (them main claims of the paper wrt. unbounded graphs. However, Reviewer #4 seems to be knowledgeable and convinced that the main results of the paper hold.

Clarity: It is difficult to follow the paper at times, as it contains some notation that is only introduced in the appendix. Furthermore, it is not always clear where to look (in the appendix) for a proof of a certain statement (e.g. Is there a proof or argumentation for Remark 1 somewhere?)

Relation to Prior Work: The paper references relevant results and gives references to proofs that are from someone else. Given that graph kernels are mentioned in the introduction, it might be interesting to cite Gaertner, Flach, Wrobel: Graph Kernels: Hardness results and efficient alternatives. which shows that computing injective graph kernels are at least as hard as computing isomorphism.

Reproducibility: Yes

Additional Feedback: Section 2 and in particular Section 3 are basically a sequence of Lemmata and Theorems with very little connecting sentences. While the paper is surprisingly readable for this being the case, it would be helpful to give the reader some more guidelines. Alg. 1 is an exponential algorithm. Alg. 2 might not be exponential. There is more going on to go from Alg. 1 to Alg. 2 than just making sure that labels are unique. This should be mentioned in more detail. Corr. 1 what is \bold{C} ?


Review 4

Summary and Contributions: Summary This paper studies learning graph functions, motivated by recovering graphical structures arising from biology, society, and finance, etc. By a simple reduction (Theorem 1), the paper observes that for some tasks it suffices to focus on functions which do not confuse non-isomorphic graphs (this paper call them iso-injective functions), which is the main theme of the paper. As another main theme, the paper also studies whether learning graph functions could be performed by neural networks. Contributions are two folds: Theory: the paper proposes a new algorithmic template, the Node Parsing Algorithm (Algorithm 2) which could be instantiated by functions that in a certain sense could be approximated by neural networks (Theorem 10). Experimental Results: The algorithm template (Algorithm 2) or its simplification (Algorithm 3) achieve state-of-the-art on four benchmarks with suitable instantiation of functions.

Strengths: Strengths The experimental results achieve state-of-the-art on four datasets, slightly beating previous algorithms.

Weaknesses: Weaknesses The theoretical contributions appear to be weak. Proofs of Theorems 1, 3, and 6 follow from basic results in undergraduate topology classes (each take efforts less than 2 minutes to proof). Ideas and contributions should not be dismissed because they are simple, if they model many situations well. However, this reader is not sure that the theoretical contribution in this paper qualifies. The non-trivial part of Theorem 10, when the graph class is countably infinite, only guarantees pointwise approximation, which is a very weak convergence (approximation) guarantee. More importantly, after unwrapping the many layers of (mostly topological) abstraction in the proof, it appears not to be explaining the ability to approximate graph functions by neural networks (partly due to the use of functions whose existence is not-so constructive). No to mention that the existence result does not say anything about the functions actually used for obtaining the experimental results on benchmarks.

Correctness: Correctness Theory: The simpler claims are correct, but this reader cannot verify the more involved claims, after not following the algorithmic presentation. Experimental Results: The experimental methodology appears to be standard.

Clarity: Clarity: The paper is pretty clear before presenting the first Algorithm 1. Presenting Algorithm 1 (Subset Parsing Algorithm) does prepare this reader a little bit for the overall structure of Algorithm 2 (Node Parsing Algorithm), but this reader is caught off-guard by the sudden introduction of many more functions without much explanations. This reader is confused by the algorithmic discussions, particularly since Algorithm 2, which uses lots of auxiliary functions (s, h, c, r) without much clear motivation. The gist appears to be aggregate information from local to global while preserving iso-injectivity, a property that the paper removed in the simplified baseline algorithm.

Relation to Prior Work: Relation to prior work: The relevant work are clearly cited and discussed, with main improvement summarized in the table.

Reproducibility: Yes

Additional Feedback: Additional feedback: - The proposal to investigate distances between isomorphism graphs sounds very reasonable, particularly to consider the sharing of subgraphs. - Section 6 Broader Impact is written in such a generic way that it could describe any papers studying machine learning or artificial intelligence.

[Author Response · NeurIPS 2020]

We thank the reviewers for their comments and feedback. We believe issues of language, terminology, and organization
can be fully addressed in the revision. We address specific issues below.

# 1 Reviewer 2

*Why is the slowness of current graph isomorphism algorithms relevant to the problem of producing isomorphism-injective*
*graph representations?*

Producing isomorphism-injective representations for graphs solves the graph canonization problem, which in turn
solves the graph isomorphism problem. Thus, the runtime of the fastest graph isomorphism algorithm is a upper bound
for the runtime of the fastest isomorphism-injective function. Thus, we focus on *multivalued functions* rather than on
*functions*.

*Page 3: the comment that the running time in algorithm is a function of the input seems odd. The run time of any*
*algorithm will be in terms of the input*

We will rephrase this comment to make it clearer. It is not the run time per se that we want to comment on, but rather
that the theorem is contingent on the *number of recursive applications* of the RNN. However, it is hard to allow the
RNN to recursively run on input for a variable number of steps. In contrast, the theorems cited for NNs as universal
function approximators are true for only one application of the NN.

*Page 4: The justification for Postulate 1 is not very clear*

We will clarify the postulate and the surrounding discussion. We mean non-induced subgraphs and the $O$ should be a $\Omega$.

Algorithm 1: what is $d_c$? And what does an extended function mean?

$d_c$ is just the dimension of the encoding. An extended function $g : \mathcal{A} \to \mathcal{B}$ of $f : A \to B$ is such that $A \subset \mathcal{A}$ with
$g|_A = f$. We will clarify this in the text.

# 2 Reviewer 3

*Algorithm 2 needs to be analyzed in detail for runtime. In particular, Corr. 1 suggests that Alg. 2 has superpolynomial*
*runtime, that there is a second paper hidden here (in case you are faster than Babai), or that the claim of Corr. 1 is*
*incorrect*

Corr. 1 is in terms of $\mathbf{C}$ which is the multivalued function variant of $C$ as per Definition 7, therefore it does not suggest
super-polynomial runtime. We will clarify this in the text.

*Regarding Definition 1 and Lemma 1*

We will clarify this in the text but with graphs of bounded size we mean bounded in terms of number of nodes, number
of edges, and that the label functions are bounded.

*I am not sure how Remark 1 results from the argumentation below Thm. 5. Furthermore, it seems unclear to me why*
*the point $p$ to which the identified subsequence converges (called $p = Alg([G]^*)$ ) would actually be in $img(Alg)$, i.e.,*
*does there exist a graph $H$ such that $p = Alg([H])$ ?*

That's correct, we do not assume that $p = Alg([G]^*)$ is in $img(Alg)$, we will change the notation to make this clearer.
We are simply deriving that convergent subsequences cannot be avoided, since $\mathcal{G}$ is infinite and the image is bounded.

# 3 Reviewer 4

*Proofs of Theorems 1, 3, and 6 follow from basic results in undergraduate topology classes*

We agree and we do not consider Theorems 1, 3, and 6 as the contributions of this paper, but rather Theorems 7, 8, 9
and 10. We invite future work to investigate other convergence than pointwise convergence.

[Meta-Review · NeurIPS 2020]

Each reviewer believes that the paper is poorly written. The reviewers, though, have agreed that (i) the problem is interesting, that (ii) the theoretical results seem to hold, and that (iii) they are interesting. On the other hand, it is not clear whether a quick revision would solve all or many of the readability issues.